# Cyclophilin A-regulated ubiquitination is critical for RIG-I-mediated antiviral immune responses

Wei Liu[1,2†], Jing Li[1†], Weinan Zheng[1], Yingli Shang[3], Zhendong Zhao[1,2], Shanshan Wang[1,2], Yuhai Bi[1], Shuang Zhang[1], Chongfeng Xu[4], Ziyuan Duan[4], Lianfeng Zhang[5], Yue L Wang[6], Zhengfan Jiang[7], Wenjun Liu[1,2*], Lei Sun[1,2*]

[1]CAS Key Laboratory of Pathogenic Microbiology and Immunology, Institute of Microbiology, Chinese Academy of Sciences, Beijing, China; [2]University of Chinese Academy of Sciences, Beijing, China; [3]College of Veterinary Medicine, Shandong Agricultural University, Tai'an, China; [4]Institute of Genetics and Developmental Biology, Chinese Academy of Sciences, Beijing, China; [5]Key Laboratory of Human Disease Comparative Medicine, Ministry of Health, Institute of Laboratory Animal Science, Chinese Academy of Medical Sciences and Comparative Medical Center, Peking Union Medical College, Beijing, China; [6]Department of Pathology, University of Chicago, Chicago, United States; [7]The Education Ministry Key Laboratory of Cell Proliferation and Differentiation, School of Life Sciences, Peking University, Beijing, China

*For correspondence: liuwj@im.
ac.cn (WJL); sunlei362@im.ac.cn
(LS)

[†]These authors contributed
equally to this work

Competing interests: The
authors declare that no
competing interests exist.

Reviewing editor: Ronald N
Germain, National Institute of
Allergy and Infectious Diseases,
United States

**Abstract** RIG-I is a key cytosolic pattern recognition receptor that interacts with MAVS to induce type I interferons (IFNs) against RNA virus infection. In this study, we found that cyclophilin A (CypA), a peptidyl-prolyl *cis/trans* isomerase, functioned as a critical positive regulator of RIG-I-mediated antiviral immune responses. Deficiency of CypA impaired RIG-I-mediated type I IFN production and promoted viral replication in human cells and mice. Upon Sendai virus infection, CypA increased the interaction between RIG-I and its E3 ubiquitin ligase TRIM25, leading to enhanced TRIM25-mediated K63-linked ubiquitination of RIG-I that facilitated recruitment of RIG-I to MAVS. In addition, CypA and TRIM25 competitively interacted with MAVS, thereby inhibiting TRIM25-induced K48-linked ubiquitination of MAVS. Taken together, our findings reveal an essential role of CypA in boosting RIG-I-mediated antiviral immune responses by controlling the ubiquitination of RIG-I and MAVS.

## Introduction

The innate immune system is the first line of defense against microbial pathogen invasion via the recognition of pathogen-associated molecular patterns (PAMPs) with the help of pattern recognition receptors (PRRs) (*Barbalat et al., 2011*; *Kawai and Akira, 2010*). Among these PRRs, RIG-I like receptors (RLRs) function as cytoplasmic RNA sensors that recognize viral RNA and activate a signaling pathway, which is essential for the production of type I interferons (IFNs) (*Kato et al., 2011*). RIG-I is required for type I IFN production in response to Sendai virus (SeV), Newcastle disease virus (NDV), influenza A virus (IAV), vesicular stomatitis virus (VSV), and Japanese encephalitis virus (JEV) (*Goubau et al., 2013*; *Kato et al., 2006*; *Loo et al., 2008*). Following ligand binding, ubiquitinated RIG-I is recruited to the mitochondria-associated membrane where it binds to MAVS (also known as

**eLife digest** In mammals, an enzyme called Cyclophilin A (CypA) is found in almost all tissues and plays important roles in many biological processes including the production of proteins and inflammation. Recent work suggests that it also plays a role in fighting virus infections.

CypA can interact directly with a protein from viruses to inhibit the virus from multiplying. Several lines of evidence indicate that CypA can also regulate virus replication by stimulating the production of molecules called type I interferons, but it is not clear how this could work.

A receptor protein called RIG-I can detect the presence of a virus and interact with another protein called MAVS to stimulate immune responses, leading to the production of type I interferons. Liu, Li et al. used human cells and mice to investigate how CypA affects this process. The experiments show that the levels of CypA in cells increase during virus infection. Cells that lack CypA produce fewer type I interferon molecules, which gives the virus more of a chance to multiply. Further experiments show that CypA alters the ubiquitin-mediated protein modification of RIG-I and MAVS.

The findings of Liu, Li et al. identify a new way in which CypA boosts immune responses during virus infections. A future challenge is to develop new drugs that regulate the protein modification of RIG-I and MAVS, which may help to treat virus infections.

IPS-1, Cardif, and VISA) to initiate innate immune signaling (*Hou et al., 2011*; *Kawai et al., 2005*; *Meylan et al., 2005*; *Seth et al., 2005*; *Xu et al., 2005*).

Cyclophilin A (CypA, encoded by *PPIA*) is a peptidyl-prolyl *cis/trans* isomerase (PPIase) that is expressed ubiquitously in all type of cells. It is the major cellular target for the immunosuppressive drug cyclosporin A (CsA) and is involved in protein folding, cell signaling, inflammation, and tumorigenesis (*Handschumacher et al., 1984*; *Lu et al., 2007*). Moreover, CypA functions as an important host factor that regulates the replication of a number of viruses, including human immunodeficiency virus type I (HIV-1), hepatitis C virus (HCV), human papillomavirus (HPV), IAV, rotavirus (RV), enterovirus-71 (EV71) virus, and infectious bursal disease virus (IBDV), which expands the role of CypA in virus infection (*Bienkowska-Haba et al., 2009*; *Chatterji et al., 2009*; *Towers et al., 2003*; *He et al., 2012*; *Liu et al., 2012b, 2009*; *Qing et al., 2014*; *Wang et al., 2015*; *Xu et al., 2010*). It has been established that CypA interacts directly with viral protein to regulate virus replication. For example, our previous studies showed that CypA-overexpressing transgenic mice exhibited resistance to influenza A virus infection (*Li et al., 2016*). We further found that CypA interacted with influenza A virus M1 protein and inhibited virus replication by accelerating ubiquitin-proteasome degradation of the M1 protein (*Liu et al., 2012b, 2009*; *Xu et al., 2010*). Yet several lines of evidence indicate that CypA can also regulate virus replication through modulating host immune responses. For instance, CypA interacted with the newly synthesized HIV-1 CA domain and subsequently activated the transcription factor IRF3 to promote the production of type I IFNs in dendritic cells (*Manel et al., 2010*). CypA inhibited RV replication by facilitating IFN-$\beta$ production (*He et al., 2012*). However, the molecular mechanism of how CypA regulates virus-mediated type I IFN production is poorly understood.

The present study indicates that CypA promotes RIG-I mediated type I IFN production and inhibits viral replication both *in vitro* and *in vivo*. We further demonstrate that CypA facilitates IFN responses through promoting K63-linked ubiquitination of RIG-I and inhibiting K48-linked ubiquitination of MAVS. Therefore, our studies identify a previously unknown mechanism that CypA promotes RIG-I-mediated type I IFN production to suppress virus replication, which adds up a new facet of CypA in host antiviral immunity.

## Results

### CypA inhibits virus replication by enhancing type I IFN production

Our previous studies have demonstrated that CypA inhibits IAV replication both *in vitro* and *in vivo* (*Li et al., 2016*; *Liu et al., 2012b, 2009*). To further investigate the impact of CypA on the

replication of other RIG-I-recognized RNA viruses, such as SeV and VSV, virus growth was monitored in shRNA-based CypA-knockdown 293T cells (293T/CypA-) and wild-type (WT) 293T cells (293T/CypA+). The hemagglutination (HA) titer of SeV and median tissue culture infective dose (TCID50) of VSV in 293T/CypA- cells was strikingly increased compared with that in 293T/CypA+ cells (*Figure 1A,B*), indicating that CypA plays an inhibitory role in the replication of SeV and VSV. To confirm the role of CypA in antiviral responses in a CypA deficient system, we purchased CypA-deficient (*Ppia*$^{-/-}$) 129 mice from Jackson Laboratory and crossed them to WT 129 mice. WT, *Ppia*$^{+/-}$ and *Ppia*$^{-/-}$ mice were identified by PCR (*Figure 1—figure supplement 1A*). The absence of *Ppia* in CypA-deficient bone marrow-derived macrophages (BMDMs) was examined by semi-quantitative PCR and Western blotting (*Figure 1—figure supplement 1B and C*). We further determined the effect of CypA on SeV replication in BMDMs from WT and *Ppia*$^{-/-}$ mice. Consistent with the results in 293T cells, we found that the mRNA expression level of SeV M gene was higher in *Ppia*$^{-/-}$ BMDMs than that in WT BMDMs (*Figure 1C*). Collectively, these data suggested that CypA inhibited the replication of RIG-I-recognized RNA viruses.

Considering RIG-I-recognized RNA viruses can trigger the RIG-I-mediated signaling pathway and promote the production of type I IFN, which in turn inhibits virus replication, we examined the effect of CypA on production of type I IFNs and interferon-stimulated genes (ISGs, such as *Ifit1*, *Ifit2*, and *Ccl5*). We performed IFN-$\beta$ promoter-driven luciferase assay in 293T cells (*Figure 1D*), and quantitative PCR (*Figure 1E,G*) and ELISA assays (*Figure 1F*) in BMDMs in response to transfected Poly (I:C) or infection of SeV, VSV or IAV with the NS1R38A/K41A mutant (IAV-mut, which induces high levels of IFN-$\beta$, [*Donelan et al., 2003*]), respectively. Absence of CypA remarkably decreased RIG-I-mediated production of type I IFNs and ISGs. We also obtained similar results with other cell types, such as human primary monocytes (*Figure 1H,I*, *Figure 1—figure supplement 2C,H,I*), 293 T cells (*Figure 1—figure supplement 2A,B*) and U937 cells (*Figure 1—figure supplement 2C–G*) triggered by SeV or VSV. Collectively, these findings suggested that CypA positively regulated expression of type I IFNs and ISGs against RIG-I-recognized RNA virus infection. We further examined the effect of CypA on RIG-I-independent signaling, such as encephalomyocarditis virus (EMCV)-triggered MDA5 pathway and herpes simplex virus type 1 (HSV-1)-triggered cGAS-STING pathway. CypA promoted IFN-$\beta$ and ISGs production and inhibited the replication of EMCV (*Figure 1—figure supplement 3A,B*), but had no impact on HSV-1-triggered cGAS-STING pathway (*Figure 1—figure supplement 3C*).

Altered host cell gene expression is a universal consequence of virus infection. When we investigated the mRNA and protein levels of CypA in BMDMs, 293T cells, U937 cells and human monocytes infected with SeV, VSV or IAV-mut, we found that CypA was highly inducible, whereas Poly (I:C) transfection or IFN-$\beta$ treatment made no difference to CypA expression, (*Figure 1—figure supplement 4A–D*), suggesting that the induction of CypA can be triggered by viruses and CypA is involved in cellular antiviral response. Taken together, upon virus infection, CypA expression was upregulated, which inhibited the replication RIG-I-recognized RNA virus by enhancing production of type I IFNs. In the following studies, we used SeV as a naturally occurring agent that strongly triggers antiviral immunity via RIG-I-mediated signaling pathway.

## CypA deficiency impairs antiviral responses *in vivo*

Having known that CypA promotes type I IFN production and inhibits the replication of RIG-I-recognized RNA virus *in vitro*, we next sought to determine CypA-mediated antiviral responses *in vivo*. In a mouse model of SeV infection, all five monitored SeV-infected *Ppia*$^{-/-}$ mice died at 9 d after infection, whereas three of five (60%) SeV-infected WT mice survived and remained healthy for the duration of the infection study (*Figure 2A*), indicating deficiency of CypA accelerated SeV infection-induced death of mice. Anatomical analysis showed that the lung indices of SeV-infected mice were increased at 7 d after infection. Notably, SeV-infected *Ppia*$^{-/-}$ mice exhibited much higher lung indices than that of WT mice (*Figure 2B*). Consistently, gross lesion of lung in *Ppia*$^{-/-}$ mice infected with SeV was severer than that in WT mice (*Figure 2C*). We further performed histopathological examination of lung, nasal turbinate, and trachea tissues at day 2, 5 and 7 post infection and found that SeV-infected *Ppia*$^{-/-}$ mice displayed severe bronchopneumonia, interstitial pneumonia, congestion in blood vessels, and dropout of the mucous epithelium, whereas SeV-infected WT mice only displayed slight bronchopneumonia and congestion in blood vessels during the infection (*Figure 2D*, *Figure 2—figure supplement 1A,B*). These data showed that tissue damage was

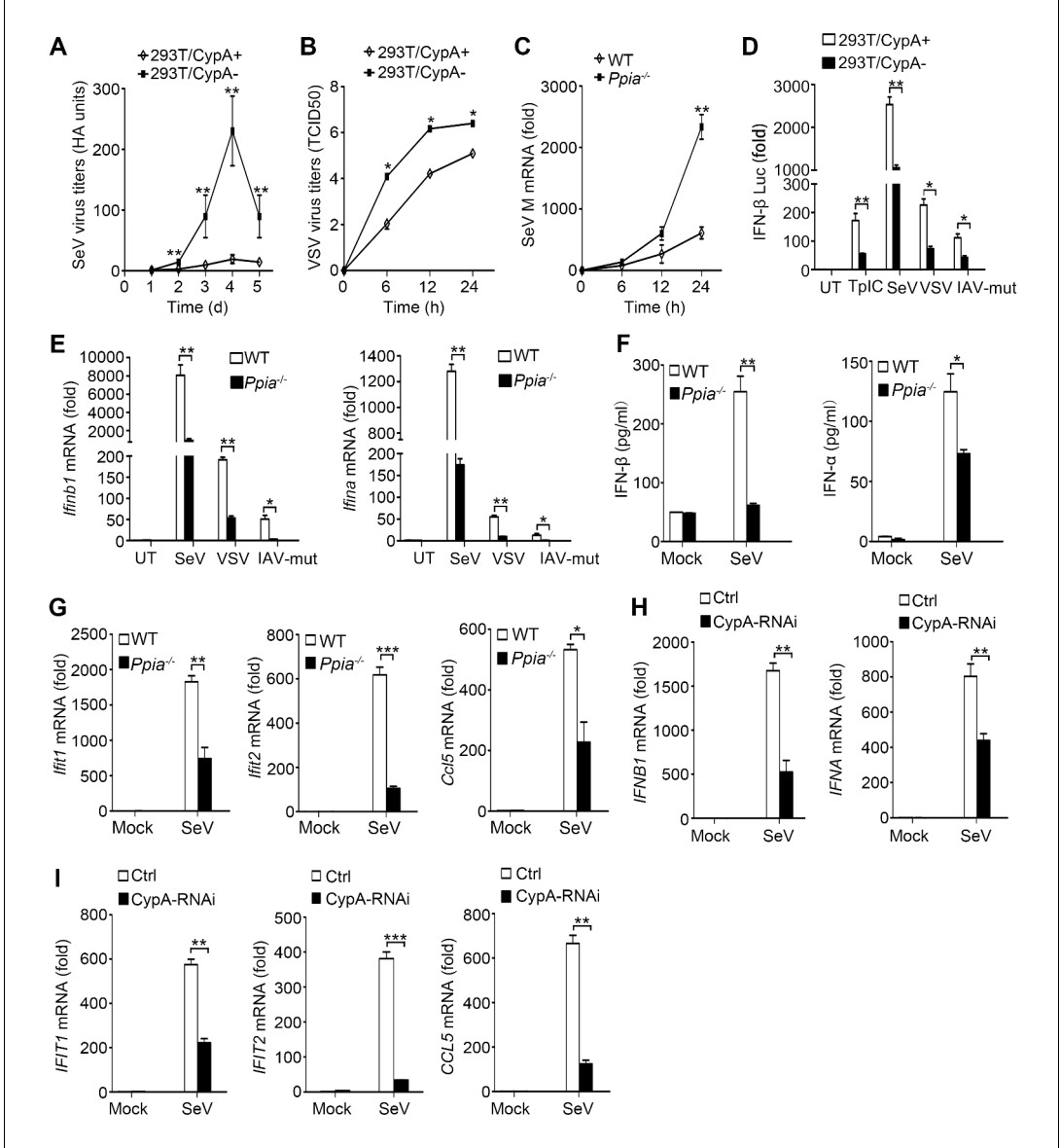

**Figure 1.** CypA promotes production of type I IFNs against virus infection. (**A**) HA assays of SeV in 293T/CypA+ or 293T/CypA- cells infected with SeV (MOI = 1) for the indicated time periods. (**B**) TCID50 assays of proliferation level of VSV in 293T/CypA+ or 293T/CypA- cells infected with VSV (MOI = 1) for the indicated time periods. (**C**) Quantitative PCR analysis of SeV M mRNA in wild-type (WT) or CypA-deficient (*Ppia$^{-/-}$*) BMDMs infected with SeV for the indicated time periods. (**D**) Luciferase activity of lysates in 293T/CypA+ or 293T/CypA- cells transfected for 24 hr with IFN-*β* luciferase reporter (IFN-*β*-Luc), together with Poly (I:C) (TpIC) or then treated with SeV, VSV, IAV-mut for 6 hr. (**E**) Quantitative PCR analysis of *Ifnb1* and *Ifna* mRNA in WT or *Ppia$^{-/-}$* BMDMs infected with SeV, VSV or IAV-mut for 6 hr. (**F**) ELISA of IFN-*β* and IFN-α production in the supernatants of WT or *Ppia$^{-/-}$* BMDMs treated with SeV for 12 hr. (**G**) Quantitative PCR analysis of *Ifit1*, *Ifit2*, and *Ccl5* mRNA in WT or *Ppia$^{-/-}$* BMDMs treated with SeV for 6 hr. (**H and I**) Quantitative PCR analysis of *IFNB1*, *IFNA* (**H**) *IFIT1*, *IFIT2*, or *CCL5* (**I**) mRNA in human monocytes transfected with CypA siRNA or scrambled siRNA for 48 hr and then treated with SeV for 6 hr. Data are shown as mean ± SD (A: n = 5; B-I: n = 3). *p<0.05, **p<0.01, ***p<0.001 (unpaired, two-tailed Student's t-test). Data are from one representative of at least three independent experiments.

The following source data and figure supplements are available for figure 1:

**Source data 1.** Quantification of viral replication and type I IFN production for *Figure 1*.

**Figure supplement 1.** Identification of *Ppia*-deficient mice.

**Figure supplement 2.** CypA promotes production of type I IFNs against virus infection in 293T, U937 cells and human monocytes.

*Figure 1 continued on next page*

*Figure 1 continued*

**Figure supplement 3.** Effect of CypA on RIG-I-independent signaling.
**Figure supplement 4.** CypA is inducible against virus infection.

aggravated in CypA deficient mice, correlating with a higher viral load in the lungs, as measured by expression of NP and M genes of SeV (*Figure 2E*). Most importantly, we found deficiency of CypA reduced expression of type I IFNs and downstream ISGs in lungs (*Figure 2F–H*) and spleens (*Figure 2—figure supplement 1C,D*), suggesting that CypA also promoted type I IFN production *in vivo*. Collectively, CypA inhibited SeV replication *in vivo* through augmenting of expression of IFNs and downstream ISGs.

## CypA promotes activation of IRF3 and NF-κB signaling pathways

It is well established that expression of type I IFN genes is mainly regulated by two transcription factors IRF3 and NF-κB. To investigate the effect of CypA on SeV-induced activation of IRF3 and NF-κB, interferon stimulated response element (ISRE) and NF-κB luciferase reporter constructs were co-transfected with CypA or control vector in 293T/CypA- cells. We found that ISRE- and NF-κB-responsive luciferase activity induced by SeV infection were dramatically lower in the absence of CypA (*Figure 3A,B*), indicating that CypA was involved in both IRF3- and NF-κB-mediated type I IFN expression. In line with this observation, the activated dimer form of IRF3 (*Figure 3C*) and phosphorylation of IRF3 and p65 (*Figure 3D*) were distinctly suppressed in SeV-infected 293T/CypA- cells compared with those in SeV-infected 293T/CypA+ cells. Moreover, CypA deficiency also inhibited phosphorylation of IRF3, p65, IKKα/β, and IκBα in cultured BMDMs, accompanied with lower expression levels of RIG-I and MAVS (*Figure 3E*). These data indicate that CypA is vital for activation of IRF3 and NF-κB signaling pathways.

## CypA targets RIG-I and MAVS to regulate RIG-I signaling pathway

In an attempt to identify the target protein of CypA, we initially tested the effect of CypA on different components of the RLR pathway in their activation of relevant promoters in a reporter assay. 293T/CypA- cells were transfected with expression vectors containing RIG-I-N (CARD domain of RIG-I), MDA5-N (CARD domain of MDA5), MAVS, TBK1, or IRF3/5D (activated form of IRF3), CypA or control vector, together with luciferase reporter constructs driven by promoters of genes encoding IFN-β, or the transcription factor NF-κB or ISRE. We found that CypA promoted the activation of the IFN-β promoter (*Figure 4A*), ISRE (*Figure 4B*) and NF-κB (*Figure 4C*) induced by overexpression of RIG-I-N, MDA5-N, and MAVS, but not induced by TBK1 or IRF3/5D, suggesting that RIG-I, MDA5, and MAVS were involved in CypA-regulated RLR pathway. Consistent with these findings, CypA increased the dimerization of IRF3 induced by overexpression of RIG-I, MDA5 and MAVS (*Figure 4D*). We next sought to determine whether CypA could interact with these key components. Coimmunoprecipitation assay did show that CypA interacted with the transfected RIG-I, MDA5, or MAVS in 293T cells (*Figure 4E*). We further observed the endogenous CypA-RIG-I interaction and CypA-MAVS interaction in SeV-infected conditions (*Figure 4F,G*). Confocal microscopy experiments indicated that CypA co-localized with the endogenous RIG-I and MAVS in 293T cells after infection with SeV (*Figure 4H*), which are consistent with the results of *Figure 4F,G*, indicating that RIG-I and MAVS were the target proteins of CypA to augment RIG-I-mediated type I IFN production.

## CypA enhances TRIM25-mediated K63-linked ubiquitination of RIG-I to facilitate recruitment of RIG-I to MAVS

Following ligand binding, RIG-I is ubiquitinated by the E3 ligase TRIM25. We have known that RIG-I is the target of CypA, it is necessary to investigate whether CypA is involved in this process. Therefore, we first investigated the effect of CypA on ubiquitination of RIG-I. In 293T/CypA- cells, transfected CypA enhanced exogenous TRIM25-mediated K63-linked, but not K48-linked ubiquitination of RIG-I (*Figure 5A*) and increased the interaction between TRIM25 and RIG-I (*Figure 5C*). Furthermore, we observed similar results of endogenous ubiquitination of RIG-I and endogenous interaction

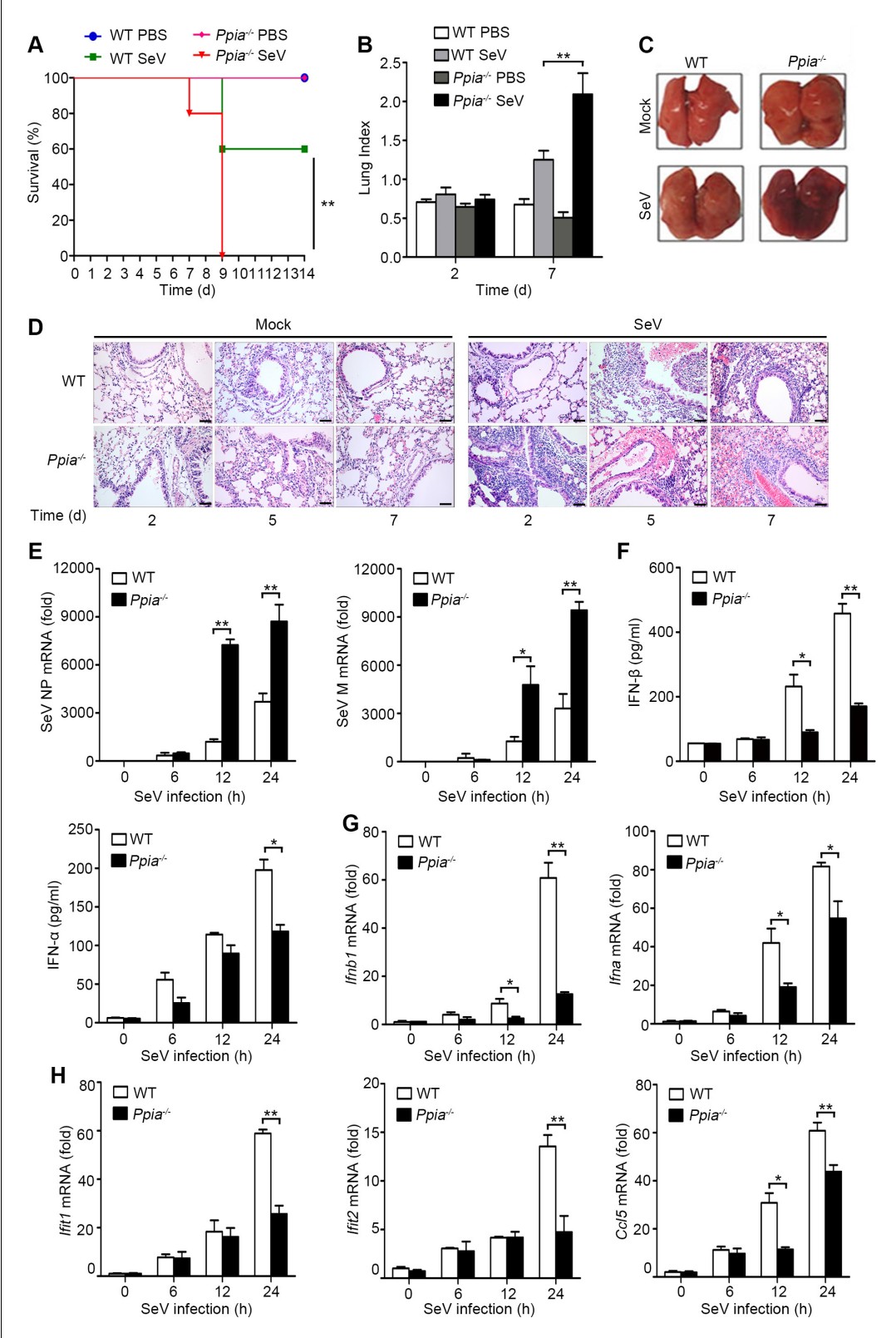

**Figure 2.** CypA positively regulates type I IFN production and antiviral responses *in vivo*. (**A**) Survival of WT and *Ppia*$^{-/-}$ mice (n = 5) infected with SeV (2000 PFU/mouse) via nasal inoculation and monitored for 14 days. (**B**) Lung index (100× lung/body weight) of WT and *Ppia*$^{-/-}$ mice (n = 5) infected with SeV for 2 and 7 days. (**C**) Gross lesion of lungs from WT and *Ppia*$^{-/-}$ mice inoculated with SeV for 7 days. (**D**) H & E stainings of lungs of WT and *Ppia*$^{-/-}$ mice (n = 3) infected with SeV or mock-infected with PBS for 2, 5, and 7 days. Scale bars, 100 μm. (**E**) Quantitative PCR analysis of SeV NP or M

*Figure 2 continued on next page*

*Figure 2 continued*

mRNA in WT or *Ppia*⁻/⁻ mice treated with SeV for the indicated time points. (**F**) ELISA of IFN-*β* and IFN-α production in lung tissues of WT or *Ppia*⁻/⁻ mice treated with SeV for the indicated time points. (**G and H**) Quantitative PCR analysis of *Ifnb1*, *Ifna* (**G**), *Ifit1*, *Ifit2,* and *Ccl5* (**H**) mRNA in lung tissues of WT or *Ppia*⁻/⁻ mice treated with SeV for the indicated time points. Data are shown as mean ± SD (B: n = 5; E-H: n = 3). *p<0.05 and **p<0.01 (unpaired, two-tailed Student's t-test). Data are representative of two independent experiments.

The following source data and figure supplement are available for figure 2:

**Source data 1.** Quantification of survival, lung index, viral replication and type I IFN production for *Figure 2*.
**Figure supplement 1.** CypA positively regulates type I IFN production and the antiviral responses *in vivo*.

between TRIM25 and RIG-I in BMDMs upon SeV infection (*Figure 5B,D*). It is well known that TRIM25 interacts with RIG-I-N to deliver the Lys 63-linked ubiquitin to the CARDs of RIG-I (*Gack et al., 2007*). We performed coimmunoprecipitation assays to explore the CypA-binding

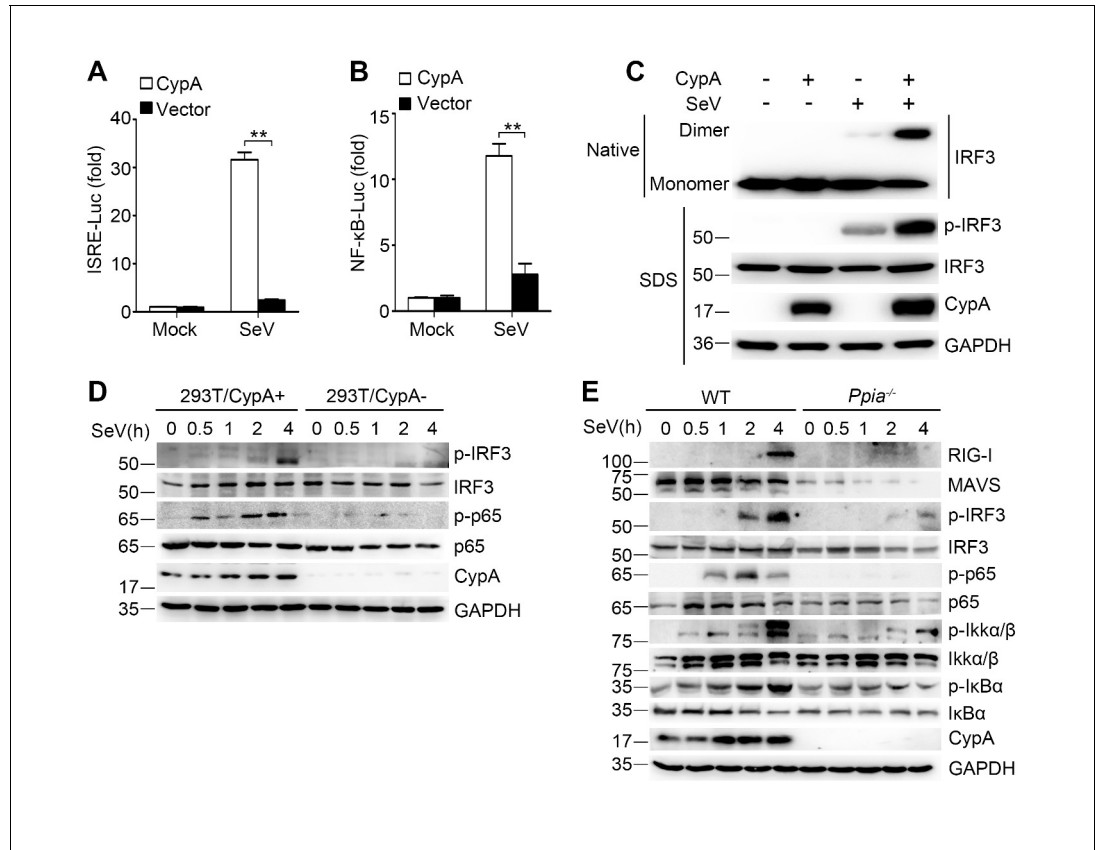

**Figure 3.** CypA deficiency suppresses IRF3 and NF-κB activation. (**A and B**) Luciferase activity of lysates in 293T/CypA- cells transfected for 24 hr with CypA or control vector, together with either ISRE-Luc (**A**) or NF-κB-Luc (**B**) and then treated with SeV for 6 hr. Results are presented relative to the luciferase activity in control cells treated with luciferase reporter and empty vector. (**C**) Native PAGE and immunoblot analysis of IRF3 in dimer or monomer form and phosphorylated IRF3 in 293T/CypA+ and 293T/CypA- cells infected with SeV for 6 hr. (**D**) Immunoblot analysis of the indicated proteins in 293T/CypA+ and 293T/CypA- cells infected with SeV for the indicated time periods. (**E**) Immunoblot analysis of the indicated proteins in WT and *Ppia*⁻/⁻ BMDMs from infected with SeV for the indicated time points. Data are shown as mean ± SD (n = 3). *p<0.05 and **p<0.01 (unpaired, two-tailed Student's t-test). Data are representative of at least three independent experiments.

The following source data is available for figure 3:

**Source data 1.** Quantification of luciferase activity for *Figure 3*.

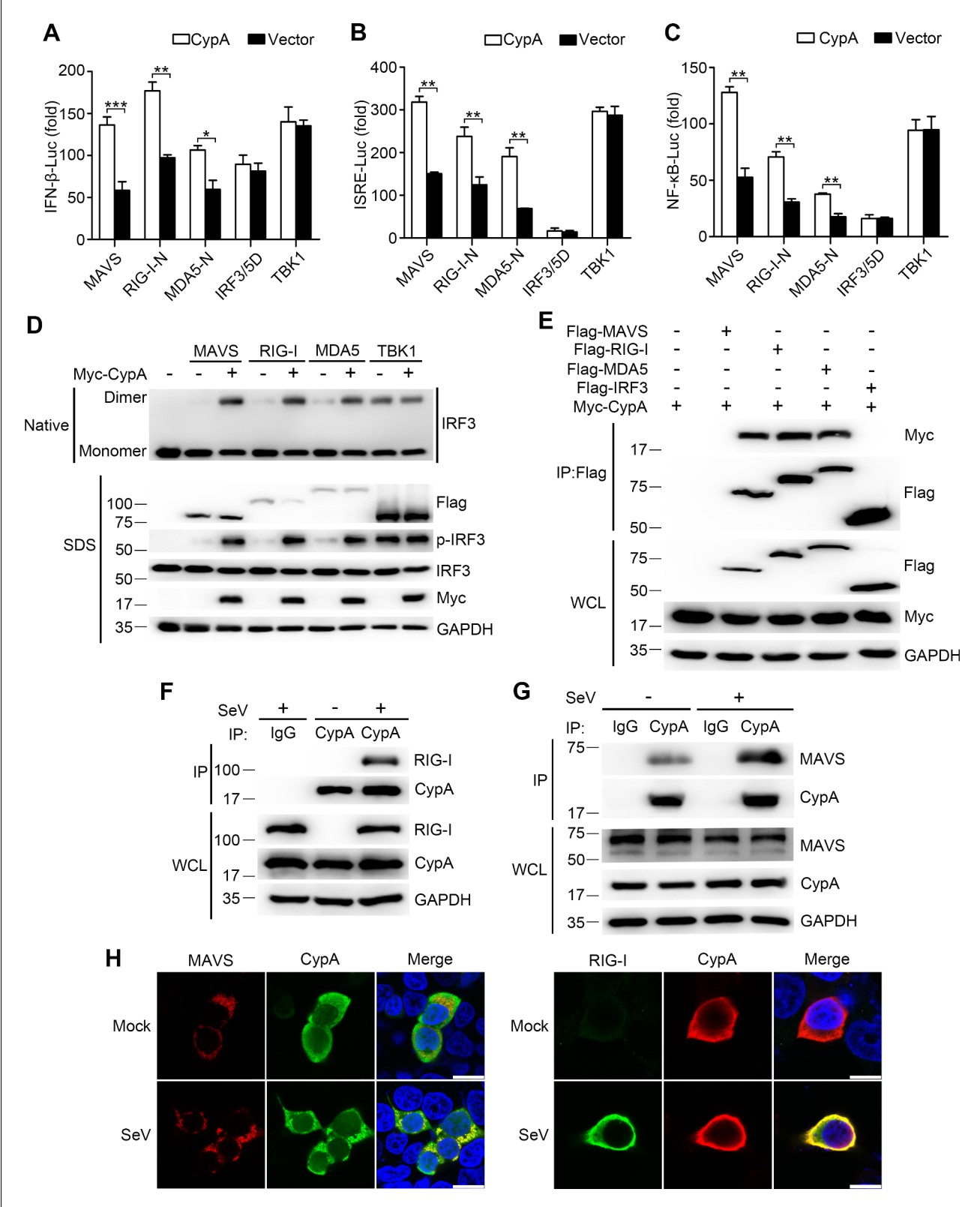

**Figure 4.** CypA interacts with RIG-I and MAVS to activate RIG-I signaling pathway. (**A–C**) Luciferase activity of lysates in 293T/CypA- cells transfected for 24 hr with luciferase reporter constructs IFN-$\beta$-Luc (**A**), ISRE-Luc (**B**) or NF-$\kappa$B-Luc (**C**), plus Flag-MAVS, Flag-RIG-I-N, Flag-MDA5-N, Flag-IRF3/5, or Myc-TBK1, along with Myc-CypA or an empty vector. Results are presented relative to the luciferase activity in control cells treated with luciferase reporter and empty vector. (**D**) Native PAGE and immunoblot analysis of IRF3 in dimer or monomer form and phosphorylated IRF3 in 293T/CypA- cells

*Figure 4 continued on next page*

*Figure 4 continued*

transfected for 24 hr with Flag-MAVS, Flag-RIG-I, Flag-MDA5, or Myc-TBK1, along with an empty vector or Myc-CypA. (E) Immunoblot analysis of lysates of 293T/CypA+ cells transfected for 24 hr with Flag-MAVS, Flag-RIG-I, Flag-MDA5, or Flag-IRF3, along with Myc-CypA, followed by immunoprecipitation with anti-Flag beads. (F and G) Immunoblot analysis of lysates in WT BMDMs infected with SeV for 6 hr, followed by immunoprecipitation with control mouse IgG or anti-CypA antibodies. Lysates and immunoprecipitation extracts were probed with CypA and RIG-I (F) or MAVS (G) antibodies. (H) Confocal microscopy of endogenous CypA and MAVS or RIG-I in 293T/CypA+ cells, treated with SeV for 6 hr. Scale bars, 10 μm. Data are shown as mean ± SD (n = 3). *p<0.05 and **p<0.01 (unpaired, two-tailed Student's t-test). Data are representative of at least three independent experiments.
The following source data is available for figure 4:

**Source data 1.** Quantification of luciferase activity for *Figure 4*.

region of RIG-I. We found that the CypA interacted with RIG-I-C (C-terminal of RIG-I) (*Figure 5E*), indicating that CypA and TRIM25 bind to different regions of RIG-I. It is possible that the binding of CypA to RIG-I facilitates the interaction between TRIM25 and RIG-I-N. We next assessed the effect of CypA on recruitment of RIG-I to mitochondria. The cytoplasmic fraction (Cyto), mitochondrial fraction (Mito) and whole-cell lysate (WCL) from SeV-infected WT and *Ppia*$^{-/-}$ BMDMs were separated for western blotting analysis. We observed that RIG-I induction was strongly decreased and much less RIG-I was found in mitochondria of BMDMs when CypA was absent (*Figure 5F*). We also detected the expression and location of endogenous RIG-I in 293T/CypA+ and 293T/CypA- cells that were stained with Mito-Tracker and infected with SeV by using confocal microscopy. The results confirmed that CypA promoted RIG-I induction and facilitated recruitment of RIG-I to mitochondria upon SeV infection (*Figure 5G*), which are consistent with the results of *Figure 5F*. We then investigated the effect of CypA on the interaction between RIG-I and MAVS. Coimmunoprecipitation experiment indicated that CypA enhanced RIG-I-MAVS interaction (*Figure 5H*). More interestingly, aside from the well-known cytoplasmic distribution (*Galat and Metcalfe, 1995*), CypA was also detected in mitochondria and its expression level was upregulated both in mitochondria and cytoplasm against SeV infection (*Figure 5F*), a finding confirmed by an immunofluorescence assay in 293T/CypA+ cells (*Figure 5I*), suggesting that CypA plays important roles in response to virus infection both in mitochondria and cytoplasm. Taken together, upon SeV infection, CypA increased the interaction between the E3 ubiquitin ligase TRIM25 and RIG-I and promoted K63-linked ubiquitination of RIG-I to facilitate recruitment of RIG-I to MAVS, leading to up-regulation of RIG-I signaling pathway.

## CypA contributes to the stability of MAVS

Notably, more MAVS was observed in WT BMDMs than in *Ppia*$^{-/-}$ BMDMs, when we assessed the protein expression levels in WCL (*Figure 3E*). Thus, we sought to examine whether CypA affects the stability of MAVS. 293T/CypA+ and 293T/CypA- cells were transfected with MAVS, RIG-I or MDA5 and treated with CHX for various times. The western blotting result showed that CypA inhibited the degradation of exogenous MAVS, but had no effect on the stability of exogenous RIG-I and MDA5 (*Figure 6A*). Also, CypA enhanced the stability of endogenous MAVS without or with SeV infection (*Figure 6B,C*). These data indicated that CypA plays key a role in stabilizing MAVS.

Proteasome- and lysosome-dependent pathways are principally responsible for intracellular protein degradation, so we investigated which pathway mediates the degradation of MAVS. In 293T/CypA+ and 293T/CypA- cells, the proteasome inhibitor MG132 significantly inhibited the degradation of MAVS, whereas the lysosome inhibitor NH$_4$Cl did not (*Figure 6D*), indicating that the degradation of MAVS is controlled by ubiquitin-meditated proteolysis. We further investigated the effects of CypA on MAVS ubiquitination. MAVS ubiquitination was inhibited in the presence of CypA (*Figure 6E*). Consistently, in coimmunoprecipitation experiments, we also found that CypA significantly decreased the ubiquitination of MAVS (*Figure 6F*). To further determine the MAVS domain responsible for CypA-regulated MAVS degradation, we tested the degradation of various MAVS truncation constructs as well as a construct of MAVS amino acids 360–540 with the substitutions K371A and K420A (360–540 KK-AA, which completely withstood MAVS degradation) (*You et al., 2009*). The absence of CypA accelerated the degradation of MAVS amino acids 360–540, while

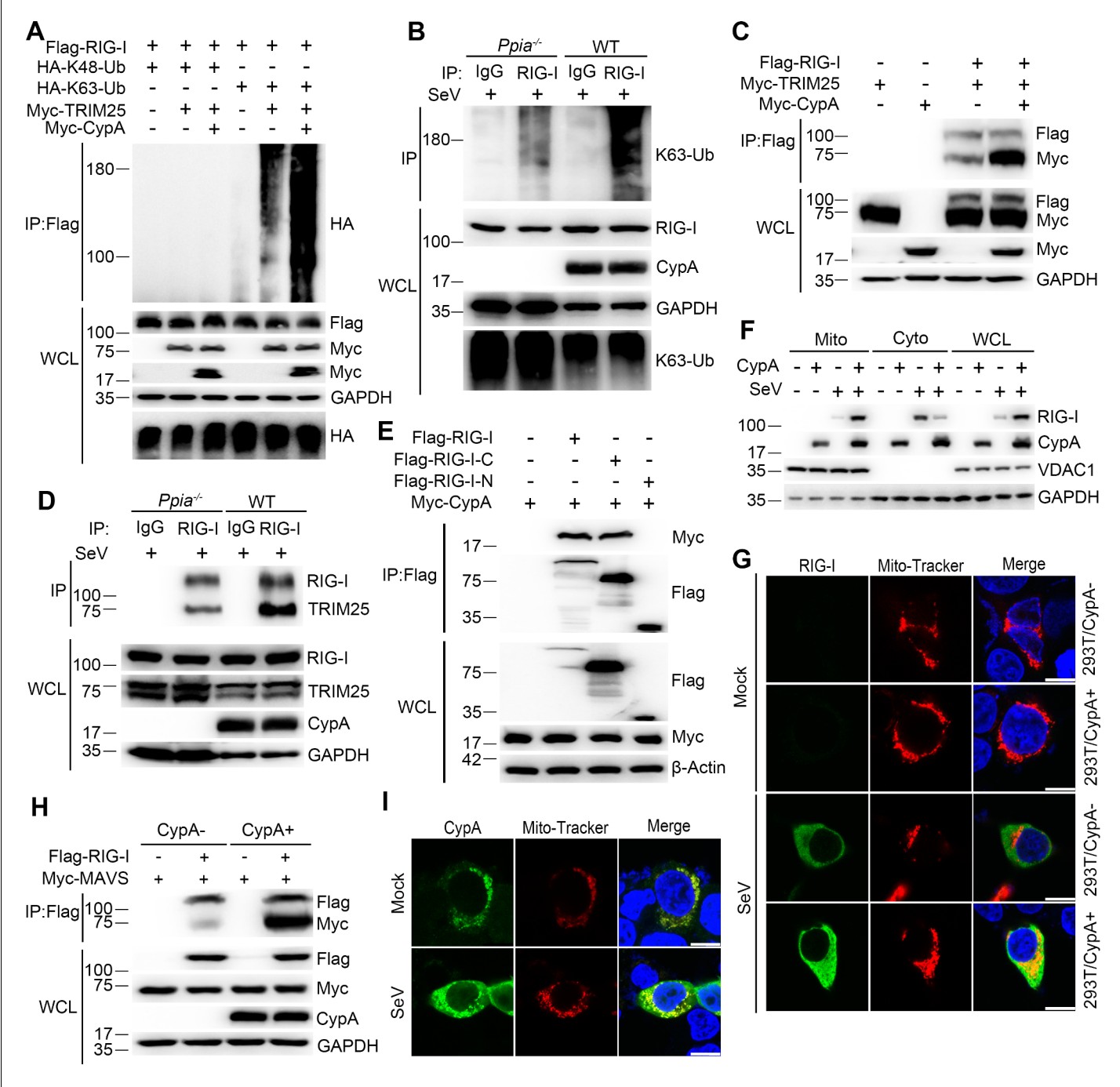

**Figure 5.** CypA enhances TRIM25-mediated K63-linked ubiquitination of RIG-I and facilitates recruitment of RIG-I to MAVS. (**A**) Immunoblot analysis of lysates in 293T/CypA- cells transfected for 24 hr with Flag-RIG-I, along with HA-K63-Ub, HA-K48-Ub, Myc-TRIM25, or Myc-CypA, followed by immunoprecipitation with anti-Flag beads. (**B**) Immunoblot analysis of lysates in WT and *Ppia*⁻/⁻ BMDMs infected with SeV for 6 hr, followed by immunoprecipitation with control mouse IgG or anti-RIG-I antibodies. Lysates and immunoprecipitation extracts were probed with K63-Ub, RIG-I and CypA antibodies. (**C**) Immunoblot analysis of lysates in 293T/CypA- cells transfected with Flag-RIG-I, Myc-TRIM25, or Myc-CypA for 24 hr, and immunoprecipitated with anti-Flag beads. (**D**) Immunoblot analysis of lysates in WT and *Ppia*⁻/⁻ BMDMs infected with SeV for 6 hr, followed by immunoprecipitation with control mouse IgG or anti-RIG-I antibodies. Lysates and immunoprecipitation extracts were probed with RIG-I, TRIM25, CypA and antibodies. (**E**) Immunoblot analysis of lysates in 293T/CypA- cells transfected with Myc-CypA and Flag-RIG-I, Flag-RIG-I-C or Flag-RIG-I-N for 24 hr, and immunoprecipitated with anti-Flag beads. (**F**) Immunoblot analysis of lysates in WT and *Ppia*⁻/⁻ BMDMs after SeV infection or mock-infection for 6 hr, followed by mitochondrial-cytoplasm extraction. (**G**) Confocal microscopy of endogenous RIG-I in 293T/CypA+ and 293T/CypA- cells stained with Mito-Tracker after SeV infection or mock infection for 6 hr. Scale bars, 10 μm. (**H**) Immunoblot analysis of lysates in 293T/CypA+ and 293T/CypA- cells

*Figure 5 continued on next page*

Figure 5 continued

transfected with Flag-RIG-I and Myc-MAVS for 24 hr, and immunoprecipitated with anti-Flag beads. (I) Confocal microscopy of endogenous CypA in 293T/CypA+ cells stained with Mito-Tracker after SeV infection or mock-infection for 6 hr. Scale bars, 10 μm. Data are representative of at least three independent experiments.

substitutions K371A and K420A completely withstood the degradation (*Figure 6G*). Together, CypA suppressed ubiquitin-mediated proteasome degradation of MAVS.

## CypA stabilizes MAVS by inhibiting TRIM25-mediated K48-linked ubiquitination of MAVS

E3 ubiquitin ligases TRIM25, Smurf1, Smurf2, AIP4, RNF5, and RNF125 have been shown to mediate K48-ubiquitination and degradation of MAVS (*Arimoto et al., 2007*; *Castanier et al., 2012*; *Pan et al., 2014*; *Wang et al., 2012*; *You et al., 2009*; *Zhong et al., 2010*), so it is important to identify the specific ubiquitin ligase involved in CypA-regulated MAVS ubiquitination. We first investigated which E3 ligase affects CypA-regulated type I IFN production. The luciferase assay indicated that CypA increased the expression of IFN-$\beta$ in the presence of TRIM25 and Smurf1 but not with AIP4, RNF125 and RNF5 (*Figure 7A*). We next evaluated whether CypA regulated TRIM25- and Smurf1-mediated MAVS stability and ubiquitination. We found that the level of endogenous MAVS protein was considerably increased in TRIM25 and CypA co-expressing cells compared with cells only transfected with TRIM25, suggesting that CypA inhibits TRIM25-mediated MAVS degradation (*Figure 7B*). In accordance with the stability assay, the ubiquitination of MAVS was inhibited in TRIM25 and CypA co-expressing cells compared with that transfected with TRIM25 (*Figure 7C*). However, CypA almost had no effect on Smurf1-mediated MAVS stability and ubiquitination (*Figure 7B,C*). Furthermore, we observed that CypA could enhance the TRIM25-mediated K48-linked, but not K63-linked ubiquitination of both exogenous and endogenous MAVS (*Figure 7D,E*).

We further determined the possible mechanism by which CypA inhibits TRIM25 mediated-MAVS ubiquitination. We speculated that CypA might compete with TRIM25 to interact with MAVS. To test this hypothesis, we performed coimmunoprecipitation experiments to analyze the interaction between TRIM25 and MAVS in the presence or absence of CypA, as well as the interaction between CypA and MAVS in the presence or absence of TRIM25. CypA-MAVS interaction was reduced in the presence of TRIM25, and TRIM25-MAVS interaction also reduced in the presence of CypA (*Figure 7F*). We next explored whether CypA and TRIM25 interacted with the same region of MAVS. Just as we speculated, CypA (*Figure 7G*) and TRIM25 (*Figure 7H*) both bind to MAVS amino acids 360–540. Thus, our results indicated that CypA and the E3 ligase TRIM25 competitively interacted with MAVS, thereby inhibiting ubiquitin-mediated proteasome degradation of MAVS.

It has been reported that TRIM25 targets MAVS at K7 and K10 for ubiquitination and AIP4 mediates MAVS ubiquitination at K371 and K420 (*Castanier et al., 2012*; *You et al., 2009*). Here we found that MAVS amino acids 360–540, which contains K371 and K420, is the common binding region for CypA and MAVS. Therefore, we investigated whether K371 and K420 were the ubiquitination sites for TRIM25-mediated MAVS ubiquitination. Both the MAVS mutation KK-AA (with the substitutions K371A and K420A) and KK-RR (with the substitutions K7R and K10R) distinctly reduced MAVS ubiquitination (*Figure 7I*), suggesting that K371 and K420, as well as K7 and K10, are the ubiquitination sites for TRIM25-mediated MAVS ubiquitination. Collectively, CypA and TRIM25 competitively interacted with MAVS, which inhibited TRIM25-mediated MAVS ubiquitination at K371 and K420.

CypA has been shown to interact with p65 and inhibit the ubiquitin-proteasome degradation of p65, thereby promoting innate immune responses (*Sun et al., 2014*). Our data showed that CypA targeted the upstream RIG-I and MAVS to upregulate RIG-I-mediated signaling pathway. To explore the relevance and contribution of the proposed CypA mechanisms at the level of RIG-I and MAVS, we tested the effect of CypA on IFN-$\beta$ induction in RIG-I-knockout cells (293T/*RIG-I*$^{-/-}$) transfected with Flag-RIG-I or Flag-MAVS, along with CypA siRNA or control siRNA. CypA increased IFN-$\beta$ expression and IRF3/p65 phosphorylation with the treatment of overexpressed RIG-I and MAVS respectively (*Figure 7J*). Then we further blocked the downstream p65 using NF-κB inhibitor (BAY-11–7082). We found that CypA still had impact on IFN-$\beta$ expression and IRF3 phosphorylation when

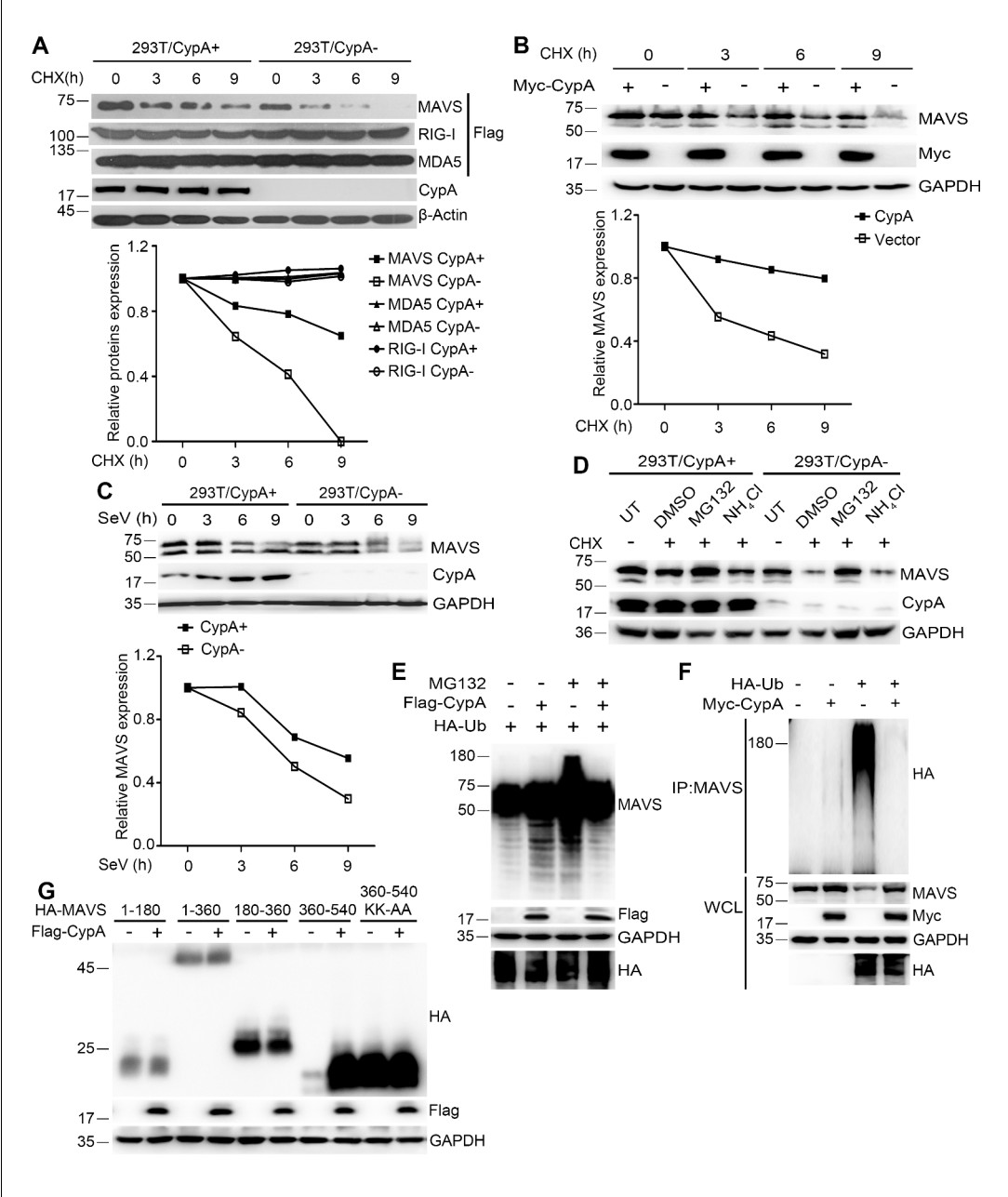

**Figure 6.** CypA suppresses ubiquitin-mediated proteasome degradation of MAVS. (**A**) Immunoblot analysis of lysates in 293T/CypA+ and 293T/CypA-cells transfected with Flag-RIG-I, Flag-MDA5 or Flag-MAVS for 24 hr and then treated with 100 μg/ml CHX for the indicated durations (top). The relative expression levels of RIG-I, MDA5 and MAVS were quantified (below). (**B**) Immunoblot analysis of lysates in 293T/CypA- cells transfected with Myc-CypA or control vector for 24 hr and then treated with 100 μg/ml CHX for the indicated time points (top). The relative expression levels MAVS were quantified (below). (**C**) Immunoblot analysis of lysates in 293T/CypA+ and 293T/CypA- cells incubated with SeV for the indicated times (top). The relative expression levels of MAVS were quantified (below). (**D**) Immunoblot analysis of lysates in 293T/CypA+ and 293T/CypA- cells treated for 6 hr with 100 μg/ml CHX, along with 10 μM NH$_4$Cl, 10 μM MG132, or DMSO. (**E**) Immunoblot analysis of lysates in 293T/CypA- cells transfected for 24 hr with HA-Ub, along with Flag-CypA or control vector and then treated with 10 μM MG132 for 6 hr. (**F**) Immunoblot analysis of lysates in 293T/CypA- cells transfected for 24 hr with HA-Ub, along with Myc-CypA or control vector and then immunoprecipitated with anti-MAVS antibody. (**G**) Immunoblot analysis of lysates in 293T/CypA- cells transfected for 24 hr with HA-tagged deletion constructs of MAVS (amino acids remaining, above lanes) and point substitution constructs containing amino acids 360–540 (KK-AA, K371A plus K420A), along with Myc-CypA or control vector. Data are representative of at least three independent experiments.

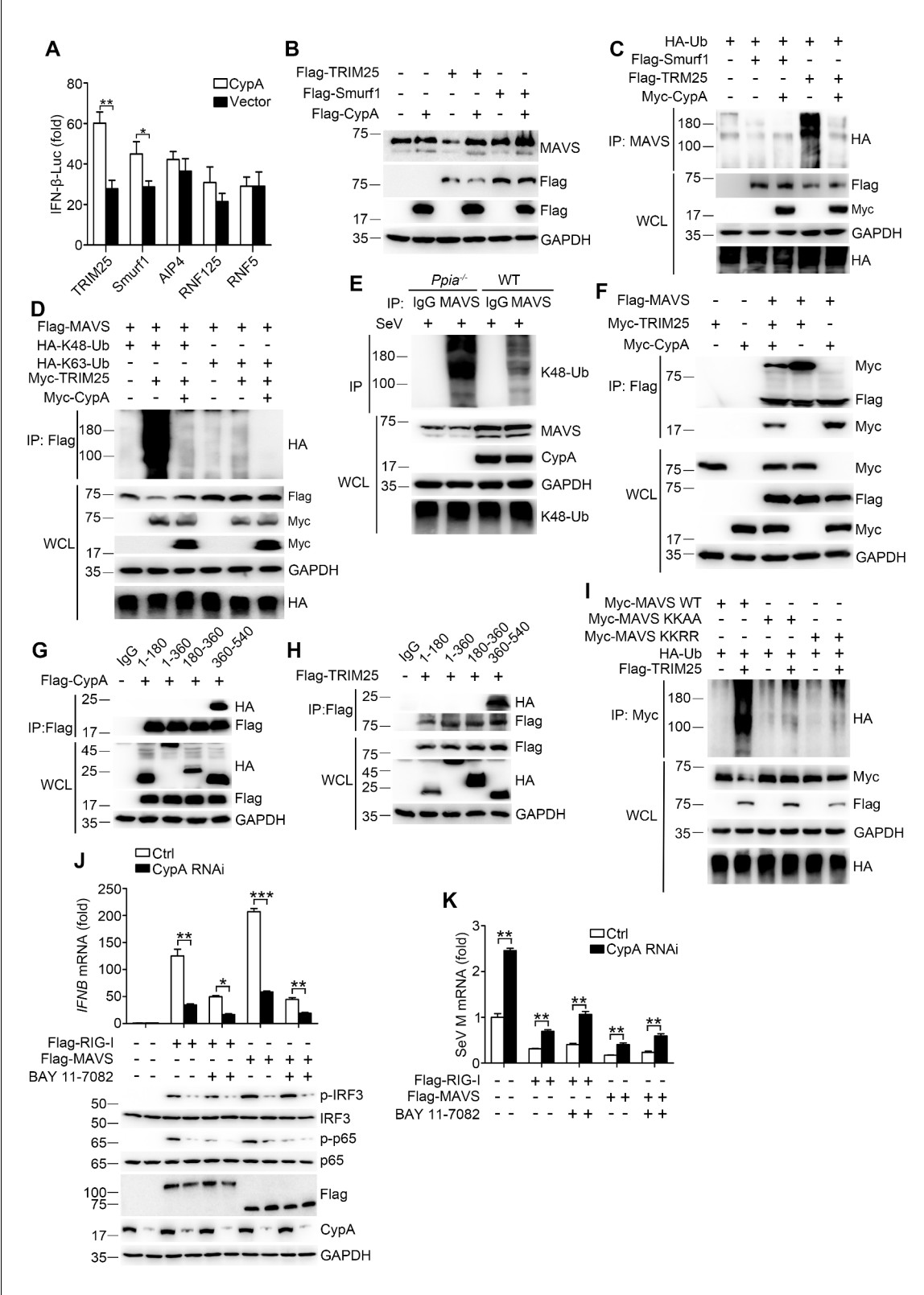

**Figure 7.** CypA inhibits TRIM25-mediated K48-linked ubiquitination of MAVS. (**A**) Luciferase activity of lysates in 293T/CypA- cells transfected for 24 hr with IFN-β-Luc and Flag-TRIM25, Flag-Smurf1, Flag-AIP4, Flag-RNF125, or Flag-RNF5, along with CypA or an empty vector and then treated with SeV for 6 hr. Results are presented relative to the luciferase activity in control cells transfected with luciferase reporter and empty vector. (**B**) Immunoblot analysis of lysates in 293T/CypA- cells transfected with various combinations of plasmids for 24 hr. (**C**) Immunoblot analysis of lysates in 293T/CypA-

*Figure 7 continued on next page*

*Figure 7 continued*

cells transfected with various combinations of plasmids for 24 hr, followed by immunoprecipitation with anti-MAVS antibody. (**D**) Immunoblot analysis of lysates in 293T/CypA- cells transfected with various combinations of plasmids for 24 hr, followed by immunoprecipitation with anti-Flag beads. (**E**) Immunoblot analysis of lysates in WT and *Ppia*$^{-/-}$ BMDMs infected with SeV for 6 hr, followed by immunoprecipitation with control mouse IgG or anti-MAVS antibodies. Lysates and immunoprecipitation extracts were probed with K48-Ub, MAVS and CypA antibodies. (**F**) Immunoblot analysis of lysates in 293T/CypA- cells transfected with various combinations of plasmids, followed by immunoprecipitation with anti-Flag beads. (**G and H**) Immunoblot analysis of lysates in 293T/CypA+ cells transfected for 24 hr with Flag-CypA (**G**) or Flag-TRIM25 (**H**), along with HA-tagged deletion constructs of MAVS, followed by immunoprecipitation with anti-Flag beads. (**I**) Immunoblot analysis of lysates in 293T/CypA+ cells transfected with HA-Ub, plus Myc-MAVS, Myc-MAVS KK-AA (K371A plus K420A), or the double point substitution construct Myc-MAVS KK-RR (K7R plus K10R), along with Flag-TRIM25 or an empty vector, followed by immunoprecipitation with anti-Myc beads. (**J**) Quantitative PCR analysis of *IFNB1* mRNA in 293T/*RIG-I*$^{-/-}$ cells pretreated for 1 hr with BAY 11–7082 (5 µM) or DMSO, and then transfected for 48 hr with Flag-RIG-I, Flag-MAVS or an empty vector, along with scrambled siRNA or CypA siRNA (top). The phosphorylated IRF3 and p65 were detected by immunoblot (below). (**K**) Quantitative PCR analysis of SeV M mRNA in 293T/*RIG-I*$^{-/-}$ cells pretreated for 1 hr with BAY 11–7082 (5 µM) or DMSO, then transfected for 48 hr with Flag-RIG-I, Flag-MAVS or an empty vector, along with scrambled siRNA or CypA siRNA, and then infected with SeV for 6 hr. Results are presented relative to mRNA level of SeV M in control cells transfected with empty vector and infected with SeV. Data are shown as mean ± SD (n = 3). *p<0.05 and **p<0.01 (unpaired, two-tailed Student's t-test). Data are representative of at least three independent experiments.

The following source data is available for figure 7:

**Source data 1.** Quantification of luciferase activity, IFN-*β* production and SeV replication for *Figure 7*.

293T/*RIG-I*$^{-/-}$ were transfected with Flag-RIG-I and treated with BAY-11–7082, or when 293T/*RIG-I*$^{-/-}$ were transfected with Flag-MAVS and treated with BAY-11–7082 (*Figure 7J*). Collectively, these data indicate that CypA is able to promote type I IFN production at the level of RIG-I and MAVS, which is independent of the downstream p65. Accordingly, CypA could inhibit SeV replication by regulating RIG-I- and MAVS-directed type I IFN production (*Figure 7K*).

## Discussion

CypA functions as either the primary intracellular target of the immunosuppressive drug CsA (*Handschumacher et al., 1984*; *Liu et al., 1991*), or as a key modulator of some biological processes (*Lu et al., 2007*). However, the roles of CypA in host antiviral immune responses are not well understood. It had been reported that CypA was highly induced in human gastric carcinoma cell line upon H9N2 influenza virus infection by using proteomics analysis (*Liu et al., 2008*). We also found that CypA was inducible in BMDMs, 293T cells, U937 cells and human monocytes infected by SeV, VSV or IAV-mut, indicating that CypA participates in cellular antiviral response. In the present study, we identified CypA as an important host factor that promotes RIG-I-mediated type I IFN production. Deficiency of CypA greatly decreases type I IFN production, which facilitates virus replication. Furthermore, CypA increased the interaction between E3 ubiquitin ligase TRIM25 and RIG-I, promoting K63-linked ubiquitination of RIG-I that facilitated recruitment of RIG-I to MAVS. Finally, CypA and TRIM25 interacted with MAVS in a competitive manner, inhibiting TRIM25-mediated K48-linked ubiquitination of MAVS at K371 and K420. Our findings demonstrated that CypA inhibited virus replication via enhancing antiviral immune responses, uncovering a different way for CypA to regulate virus infection.

The ubiquitin system has been certified to play an essential role in precisely controlling RIG-I-mediated signal transduction (*Heaton et al., 2016*; *Maelfait and Beyaert, 2012*). Upon RNA virus infection, viral RNA bind to RIG-I followed by the binding of TRIM25 to deliver the K63 ubiquitin chains to RIG-I (*Gack et al., 2007*; *Jiang et al., 2012*; *Kowalinski et al., 2011*; *Peisley et al., 2014*; *Sanchez et al., 2016*), then the CARD domains of RIG-I are exposed to interact with the CARD domains of MAVS. In the present study, we found that CypA increased the interaction between RIG-I and TRIM25, which facilitated K63-linked ubiquitination of RIG-I and recruitment of RIG-I from the cytosol to mitochondrion-associated MAVS, suggesting a positive role of CypA in the production of type I IFN. It has also been reported that another RIG-I binding protein, 14-3-3ε, is essential for the stable interaction of RIG-I with TRIM25, which facilitates RIG-I ubiquitination and initiation of innate immunity against hepatitis C virus and other pathogenic RNA viruses. (*Liu et al., 2012a*).

Therefore, just like 14-3-3ε, CypA could be defined as a key mitochondrial targeting chaperone protein that is required for innate antiviral responses.

It has been established that MAVS undergoes K48-linked ubiquitination during virus infection, which mediates MAVS degradation (*Liu et al., 2013*; *Paz et al., 2009*). A number of studies have showed that some host proteins, including PCBP2, IRTKS and Ndfip1, promoted E3 ligase AIP4- or Smurf-mediated MAVS ubiquitination for degradation, leading to suppressed type I IFN production (*Wang et al., 2012*; *Xia et al., 2015*; *You et al., 2009*). Here we found that CypA inhibited TRIM25-mediated K48-linked ubiquitination of MAVS to slow down the degradation of MAVS, thereby facilitating RIG-I-mediated type I IFN production. All these data indicated that inhibition of MAVS degradation facilitated RIG-I-mediated type I IFN production. However, a previous study suggested that the proteasomal degradation of MAVS was required to release the signaling complex into the cytosol for phosphorylation of IRF3 and subsequent production of IFN-$\beta$ (*Castanier et al., 2012*). In that study, MG132 was used as an inhibitor of MAVS degradation to test the effect of MAVS stability on IFN production, while MG132 is not the specific inhibitor of MAVS degradation. This is a tempting and interesting hypothesis, which remains to be further studied (*Jacobs and Coyne, 2013*).

Besides CypA-regulated ubiquitination of RIG-I and MAVS, we have reported that CypA accelerates ubiquitin-proteasome degradation of the M1 protein of influenza virus and then restricts virus replication(*Liu et al., 2012b*). Additionally, CypA and another PPIase, Pin1, both enhance the stability of P65 by blocking the ubiquitin-proteasome degradation, and SOCS-1 is the ubiquitin ligase for P65 (*Ryo et al., 2003*; *Sun et al., 2014*). These results support the hypothesis that PPIases have some common function in regulating ubiquitination of proteins. Together, we reveal a ubiquitination-based mechanism by which CypA controls RIG-I-mediated antiviral immune responses.

CypA is widely distributed in almost all tissues. Multiple lines of evidence have revealed that CypA interacts with a large number of proteins and plays various biological roles through different mechanisms. We found that CypA-MAVS interaction was reduced in the presence of TRIM25, and TRIM25-MAVS interaction also appeared to be reduced in the presence of CypA. In addition, both CypA and TRIM25 interacted with a similar stretch of MAVS (aa 360–450) and within this region K371 and K420 were the ubiquitination target sites for TRIM25 as well as the sites that CypA targeted to stabilize MAVS. All these results suggest that CypA competes with TRIM25 for MAVS binding to inhibit TRIM25-mediated K48-linked ubiquitination of MAVS. But on the other hand, we found that CypA promoted the interaction between TRIM25 and RIG-I, which is a quite different mechanism from that at the level of MAVS. As is well known, TRIM25 interacts with RIG-I-N to deliver the Lys 63-linked ubiquitin to the CARDs of RIG-I (*Gack et al., 2007*). We observed that CypA interacted with RIG-I-C, indicating that CypA and TRIM25 bind to different regions of RIG-I. We speculated that the conformation of RIG-I-N might be changed as soon as CypA interacted with RIG-I-C, then the binding site of TRIM25 was exposed, which facilitated the interaction between TRIM25 and RIG-I-N. A detailed structural study is an interesting future direction.

In conclusion, our data demonstrated that CypA regulates RIG-I signaling in two ways. On one hand, CypA promotes K63-linked ubiquitination of RIG-I and recruits more RIG-I to MAVS. On the other hand, CypA stabilizes MAVS by suppressing its ubiquitin-mediated proteasome degradation. Hence, our data further expand the biological functions of CypA in RIG-I-mediated antiviral innate immunity and provide a potential novel target for manipulating viral infection.

# Materials and methods

## Cell lines and antibodies

ShRNA-based knockdown of CypA in human embryonic kidney 293T cells (CRL-3216, ATCC) has been described (*Liu et al., 2012b*). CRISPR/Cas9-based knockout of *RIG-I* in 293T cells has been described (*Jiang et al., 2016*). The cell lines were authenticated by immunoblotting with multiple markers and tested for mycoplasma contamination using the MycoAlert Mycoplasma Detection Kit (Lonza, Switzerland). The 293T cells were maintained in Dulbecco's modified Eagle's medium (GIBCO) supplemented with 10% heat-inactivated fetal bovine serum (FBS, GIBCO). U937cells (CRL-1593.2, ATCC) were maintained in 1640 medium (GIBCO) with FBS. BMDMs from 129 mice were maintained in 1640 medium (GIBCO) with FBS and maintained in macrophage-colony stimulating factor (M-CSF, 20 ng/ml) for 5–7 d. Peripheral blood was obtained from healthy donors under clinical

protocols. Human peripheral blood mononuclear cells (PBMCs) were isolated using Hypaque-Ficoll density gradients by standard techniques. Monocytes were also isolated by elutriation of leukopheresis product and repurifcation using the autoMACS system if needed. Resulting cell preparations were analyzed by staining with CD14 antibodies and analyzed on a BD LSR II system. Monocyte preparations were ≥95% CD14+. For immunoblot analysis, the following antibodies were used: rabbit polyclonal antibodies to human CypA were generated as previously described, (1:2000, [*Liu et al., 2009*]), anti-c-Myc (1:2000, C3956, Sigma, RRID:AB_439680), anti-FLAG M2 (1:2000, F3165, Sigma, RRID:AB_259529), anti-$\beta$-actin (1:1000, sc-1616-R, Santa Cruz, RRID:AB_630836), anti-GAPDH (1:1000, sc-25778, Santa Cruz, RRID:AB_10167668), anti-MAVS (1:1000, sc-68881 and sc-166583, Santa Cruz, RRID:AB_1565328 and AB_2012300), anti-IRF3 (1:1000, sc-9082, Santa Cruz, RRID:AB_2264929), anti-mouse IgG (1:1000, sc-137075, Santa Cruz, RRID:AB_2285870), anti-p-IRF3 (Ser396) (1:1000, 4947, CST, RRID:AB_823547), anti-p65 (1:1000, 8242, CST, RRID:AB_10859369), anti-p-p65 (Ser536) (1:1000, 3033, CST, RRID:AB_331284), anti-IKKα/β (1:1000, 2682 and 2370, CST, RRID:AB_331626 and AB_2122154), anti-p-IKKα/β (Ser176/180) (1:1000, 2697, CST, RRID:AB_2079382), anti-IκBα (1:1000, 4814, CST, RRID:AB_390781), anti-p-IκBα (Ser32) (1:1000, 2859, CST, RRID:AB_561111), anti-RIG-I (1:1000, 3743, CST, RRID:AB_2269233), and anti-MDA5 (1:1000, 5321, CST, RRID:AB_10694490), anti-VDAC (1:1000, ab14734, Abcam, RRID:AB_443084), anti-TRIM25 (1:1000, 12573–1-AP, Proteintech, RRID:AB_2209732), anti-Ubiquitin Antibody, Lys48-Specific (1:1000, 05–1307, Millipore, RRID:AB_1587578), anti-Ubiquitin Antibody, Lys63-Specific (1:1000, 05–1308, Millipore, RRID:AB_1587580). For immunofluorescence analysis, the following antibodies were used: CypA (1:100, [*Liu et al., 2009*]), anti-MAVS (1:50, 3993, CST, RRID:AB_823565), and anti-RIG-I (1:50, MABF297, Millipore, RRID:AB_2650546).

## Plasmids

The IFN-$\beta$ promoter luciferase reporter plasmid (IFN-$\beta$-Luc) and NF-κB promoter luciferase reporter plasmid (NF-κB-Luc) were provided by C. Zheng (Su Zhou University, China). The ISRE-promoter luciferase reporter plasmid (ISRE-Luc), TBK1 and MAVS expression plasmids were provided by R. Lin (McGill University, Canada). The RIG-I-N expression plasmid was provided by T. Fujita (Tokyo Metropolitan Institute of Medical Science, Japan). RIG-I-C was synthesized by GENEWIZ and then cloned into pcDNA3.0-Flag vector. The MDA5-N expression plasmid was provided by S. Goodbourn (University of London, United Kingdom). The IRF3/5D expression plasmid was provided by Y. Lin (National Defense Medical Center, Taiwan). HA-tagged deletion constructs of MAVS or point substitution constructs containing amino acids 360–540 KKAA (K371A plus K420A) and AIP4, expression plasmids were provided by Z, Jiang (Peking University, China). MAVS KKAA (K371A plus K420A), MAVS KKRR (K7R plus K10R) were synthesized by GENEWIZ and cloned into pCMV-myc vector respectively. The HA-Ub, HA-K48-Ub, HA-K63-Ub, TRIM25, Smurf1, RNF125 and RNF5 expression plasmids were provided by X. Ye (Chinese Academy of Sciences, China).

## CypA-deficient mice

CypA-deficient ($Ppia^{-/-}$) 129 mice were purchased from Jackson Laboratory and crossed to WT 129 mice. For analysis of the genotype of each mouse, genomic DNA was isolated from tail tissue and was identified by PCR using the primers $Ppia$-oIMR3772, 5'-GCAGTTGTGATTGATCCAGGTCCG-3'; $Ppia$-oIMR3773, 5'CACCCTGGAGCACCACTGCCCACC-3'; and $Ppia$-oIMR3774, 5'-CCTGATCGA-CAAGACCGGCTTCC-3'. The animal research was approved by the Research Ethics Committee of Chinese Academy of Sciences (Permit Number: PZIMCAS2013001), and complied with the Beijing Laboratory Animal Welfare and Ethical Guidelines of the Beijing Administration Committee of Laboratory Animals.

## SeV infection

For virus infection of cells, the culture medium was removed from the plates, and the cells were washed twice with PBS. Serum-free culture medium containing SeV (MOI = 1) was added for 2 hr, and then the old medium was replaced with 2% FBS culture medium. For virus infection in 8-week-old WT and $Ppia^{-/-}$ mice, SeV (2000 PFU/mouse) was intranasally injected into the mice. The day of virus inoculation was defined as day 0. Mice were killed at 2, 5, or 7 d after infection, and the lung indices (100× lung/body weight) were measured. Lung tissues were then fixed, sectioned at 5 μm

and stained with hematoxylin and eosin. For cytokines and SeV replication analyses, mice were killed at 6, 12, and 24 hr after infection. Lung tissues were homogenized using a QIAGEN Tissue Lyser II machine (30 cycles/s, 4 min) in 1 ml of cold PBS under sterile conditions. Total RNA was extracted from homogenized lung tissue using TRIzol (Invitrogen) to detect the mRNA level of cytokines and viral genes. Then, the remaining homogenates were centrifuged and the supernatants were used to detect the protein levels of IFN-$\beta$ and IFN-$\alpha$ *via* ELISA (PBL Assay Science).

## Hemagglutination (HA) assay

SeV virus was harvested from the supernatants of infected cells every day. A standardized concentration of chicken red blood cells (0.5% RBC) was used. A serial twofold dilution of supernatant was prepared in U-bottomed 96-well microtiter plates with PBS, 50 µl 0.5% RBC was added to each well, and the U-bottomed plates were incubated for 30 min at room temperature. Then, the lattice forming parts were counted, and the titer was calculated.

## VSV titration

MDBK cells were seeded in 96-well plates 24 hr before VSV virus infection. VSV virus supernatants were serially diluted with DMEM and added to each well with eight replicates of each dilution. 24 hr after infection, the 50% TCID50 was calculated by the Reed-Muench method.

## Luciferase assay

293T cells were seeded into 24-well plates. The following day, cells were transfected with 200 ng luciferase plasmid and 100 ng $\beta$-Gal plasmid, along with 200 ng to 400 ng plasmids required for different experiments. Twenty-four hours later, cells were lysed in lysis buffer. After centrifugation, the supernatants were stored at $-80°C$. The luciferase assays were performed with a luciferase assay kit (Promega, Madison, WI).

## RNA extraction, cDNA synthesis, and Quantitative PCR analysis

Total RNA was extracted from cells with TRIzol (Invitrogen) according to the manufacturer's instructions. Samples were digested with DNase I and subjected to reverse transcription-PCR (RT-PCR). RNA was reverse-transcribed using an oligo (dT) primer. A mock reaction was performed with no reverse transcriptase added. The analysis of the relative gene expression levels was performed using Corbett 6200 and PCR primers: hIFN-$\beta$ (*IFNB1*) forward, 5'-AACTGCAACCTTTCGAAGCC-3'; hIFN-$\beta$ (*IFNB1*) reverse, 5'-TGTCGCCTACTACCTGTTGTGC-3'; mIFN-$\beta$ (*Ifnb1*) forward, 5'-GGAGATGACG-GAGAAGATGC-3'; mIFN-$\beta$ (*Ifnb1*) reverse, 5'-CCCAGTGCTGGAGAAATTGT-3'; mIFN-$\alpha$ (*Ifna*) forward, 5'-GGCTTGACACTCCTGGTACAAATGAG-3'; mIFN-$\alpha$ (*Ifna*) reverse, 5'-CAGCACA TTGGCAGAGGAAGACAG-3'; hISG54 (*IFIT2*) forward, 5'-TCATTTTGCATCCCATAGGAGGTT-3'; hISG54 (*IFIT2*) reverse, 5'-GACTTTGGTCCCCCAGCTTT-3'; mISG54 (*Ifit2*) forward, 5'-ATGAA-GACGGTGCTGAATACTAGTGA-3'; mISG54 (*Ifit2*) reverse, 5'-TGAGGGCTTTCTTTTTCC-3'; hISG56 (*IFIT1*) forward, 5'-TTCGGAGAAAGGCATTAGA-3'; hISG56 (*IFIT1*) reverse, 5'-TCCAGGGCTTCA TTCATAT-3'; mISG56 (*Ifit1*) forward, 5'-CAGAAGCAC ACATTGAAGAAGC-3'; mISG56 (*Ifit1*) reverse, 5'-TGTAAGTAGCCAGAGGAAGGTG-3'; hRantes (*CCL5*) forward, 5'-TGCCTGTTTCTGCTTGCTC TTGTC-3'; hRantes (*CCL5*) reverse, 5'-TGTGGTAGAATCTGGGCCCTTCAA-3'; mRantes (*Ccl5*) forward, 5'-ACTCCCTGCTGCTTTGCCTAC-3'; mRantes (*Ccl5*) reverse, 5'-ACTTGCTGGTGTAGAAA TACT-3'; hCypA (*PPIA*) forward, 5'- CAACCCCACCGTGTTCTTC-3'; hCypA (*PPIA*) reverse, 5'- AC TTGCCACCAGTGCCATTA-3'; mCypA (*Ppia*) forward, 5'-TTTGCAGACGCCACTGTC-3'; mCypA (*Ppia*) reverse, 5'-CAGTGCTCAGAGCTCGAAAG-3'; SeV NP forward, 5'- caagagcccactcttccaggg-3'; SeV NP reverse, 5'-CTGAACGCCTCTAACCTGTTG-3'; SeV M forward, 5'-GTGATTTGGGCGGCATC T-3'; and SeV M reverse, 5'-GATGGCCGGTTGGAACAC-3'. GAPDH served as an internal control using PCR primers, GAPDH forward 5'-TTGTCTCCTGCGACTTCAACAG-3' and GAPDH reverse 5'-GGTCTGGGATGGAAATTGTGAG-3'.

## ELISA-based IFN determination

Supernatants from cultured cells or sera were collected at the indicated times. Cytokines were analyzed by ELISA kits (Thermo) according to the manufacturer's instructions.

## Coimmunoprecipitation, immunoblot analysis and native PAGE

Cells were lysed in lysis buffer containing 0.5% NP40, 150 mM NaCl, 20 mM HEPES (pH 7.4), 10% glycerol, 1 mM EDTA, and protease inhibitor cocktail. After centrifugation, the supernatants were incubated with anti-FLAG, anti-Myc beads (Sigma), or Protein A/G PLUS-Agarose beads (Santa Cruz) for 4 hr at 4°C. After five washes in washing buffer (0.5% NP40, 300 mM NaCl, 20 mM HEPES (pH 7.4), 10% glycerol, and 1 mM EDTA), the immunoprecipitates were analyzed by immunoblot analysis. Native PAGE was performed with an 8% acrylamide gel without SDS. The gel was pre-run for 30 min at 40 mA on ice with 25 mM Tris-HCl (pH 8.4) and 192 mM glycine with or without 0.5% deoxycholate in the cathode chamber and anode chamber, respectively. Samples in the native sample buffer (50 mM Tris-HCl, pH 6.8, and 15% glycerol) were applied on the gel and underwent electrophoresis for 60 min at 35 mA on ice followed by immunoblot analysis.

## RNA interfering

Duplexes of CypA siRNA and negative controls were synthesized by Genepharma (Shanghai, China). SiRNA oligonucleotides are as follows: CypA sense, 5'-GCUCGCAGUAUCCUAGAAUTT-3'; CypA antisense, 5'-AUUCUAGGAUACUGCGAGCTT-3'; negative control sense, 5'-UUCUCCGAACGUG UCACGUTT-3'; negative control antisense, 5'-acgugacacguucggagaaTT-3'. Transfection of siRNA into cells was performed according to manufacturer's instructions. U937 cells and human monocytes were transfected with siRNA using Lipofectamine 2000 (Invitrogen).

## Indirect immunofluorescence

Cells were washed with PBS three times, fixed in 4% paraformaldehyde for 30 min at room temperature, permeabilized with 0.5% Triton X-100 in PBS (PBST) for 20 min, and stained with appropriate antibodies. Cell nuclei were stained with 5 µg/ml DAPI (Sigma). Following staining, cover slips were analyzed using a Leica SP8 confocal microscope.

## Subcellular fractionation

Cells ($5 \times 10^7$) infected with SeV or left uninfected were washed with PBS and lysed by douncing 35 times in 1.5 ml homogenization buffer (ApplyGen). The homogenates were then centrifuged at 800 g for 5 min twice. The supernatants were centrifuged at 12,000 g for 10 min to precipitate mitochondria. The supernatants from this step (cytoplasm fraction) were also collected. The precipitate fraction was washed with 0.2 ml homogenization buffer, centrifuged at 12,000 g for 10 min and collected as the mitochondria fraction.

## CHX, MG132, and NH4Cl treatment

Cells were treated with 100 µg/ml CHX for various periods of time at 24 hr after transfection. Then, cells were lysed and analyzed by immunoblotting. MG132 (10 µM) and $NH_4Cl$ (10 µM) were used at the same time as CHX, and cells were harvested 6 hr after treatment.

## Acknowledgements

This work was supported by grants from the National Natural Science Foundation of China (31472178, 31672531, and 31630079), the Key Research Program of the Chinese Academy of Sciences (KSZD-EW-Z-005–001), and the National Key Technology Support Program (2015BAD11B02). W. J.L. is the principal investigator of the Innovative Research Group of National Natural Science Foundation of China ( 81621091).

## Additional information

### Funding

| Funder | Grant reference number | Author |
| --- | --- | --- |
| National Natural Science Foundation of China | 31472178 | Lei Sun |
| National Natural Science | 31672531 | Lei Sun |

Foundation of China

| Key Research Program of Chinese Academy of Sciences | KSZD-EW-Z-005-001 | Wenjun Liu |
|---|---|---|
| National Natural Science Foundation of China | 31630079 | Wenjun Liu |
| National Key Technology Support Program of China | 2015BAD11B02 | Lei Sun |
| National Natural Science Foundation of China | 81621091 | Wenjun Liu |

The funders had no role in study design, data collection and interpretation, or the decision to submit the work for publication.

## Author contributions

WL, Data curation, Formal analysis, Methodology, Writing—original draft; JL, Formal analysis, Methodology; WZ, Formal analysis, Methodology ; YS, Formal analysis, Methodology, Writing—review and editing; ZZ, SW, YB, SZ, CX, LZ, Methodology; ZD, Formal analysis, Writing—review and editing; YLW, ZJ, Methodology, Writing—review and editing; WJL, Conceptualization, Formal analysis, Supervision, Funding acquisition, Validation, Project administration, Writing—review and editing; LS, Conceptualization, Data curation, Formal analysis, Supervision, Funding acquisition, Validation, Writing—original draft, Project administration, Writing—review and editing

## Author ORCIDs

Wei Liu, http://orcid.org/0000-0003-0004-2713
Lei Sun, http://orcid.org/0000-0003-0141-2093

## Ethics

Animal experimentation: The animal research was approved by the Research Ethics Committee of Chinese Academy of Sciences (Permit Number: PZIMCAS2013001), and complied with the Beijing Laboratory Animal Welfare and Ethical Guidelines of the Beijing Administration Committee of Laboratory Animals.

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
