## [Decision Letter]

Thank you for submitting your article "Cyclophilin A-regulated ubiquitination is critical for RIG-I-mediated antiviral immune responses" for consideration by *eLife*. Your article has been reviewed by two peer reviewers, and the evaluation has been overseen by a Reviewing Editor and Michel Nussenzweig as the Senior Editor. The reviewers have opted to remain anonymous.

The reviewers have discussed the reviews with one another and the Reviewing Editor has drafted this letter to direct your attention to key issues we feel must be considered in our decision concerning this work. We ask that you respond with details of how you would propose to address the major points specified below and an approximate timetable for the completion of his work. The editor and reviewers will consider your response and issue a recommended action.

In this manuscript Liu et al. provide data that show that cells deficient in Cyclophilin A (CypA) are impaired in type I interferon (IFN) production in response to viruses that are sensed by the intracellular RNA receptor RIG-I. The authors further demonstrate that CypA deficiency leads to increased replication of these viruses. The authors tested the interaction of CypA with RIG-I, the related sensor MDA5, and their downstream adaptor MAVS, and found that CypA binds to all three proteins. The authors provide data suggesting two apparently independent mechanisms by which CypA promotes the RIG-I-MAVS-mediated antiviral response: a) CypA promoting the interaction of the E3 ligase TRIM25 to RIG-I and subsequent K63-linked ubiquitination of RIG-I by TRIM25; and b) CypA competing with TRIM25 for MAVS binding, thereby blocking the degradative ubiquitination of MAVS by TRIM25.

Strengths of this study are the phenotypical data that demonstrate that CypA modulates the IFN-mediated antiviral response. A major weakness of the study is that the proposed mechanism(s) by which CypA facilitates IFN-mediated innate immunity is not well supported by the data, and that many of the biochemical experiments are poorly executed.

1) The authors propose two independent mechanisms by which CypA promotes RIG-I-MAVS signaling but fail to integrate these two mechanisms, nor demonstrate the relevance and contribution of each mechanism to CypA-mediated IFN induction. This is especially important as CypA has been shown to act also downstream of RLRs-MAVS (as cited by the authors in the Discussion section). As such, the authors need to demonstrate the relevance of their proposed mechanisms for IFN induction and viral replication.

2) The authors provide data showing that CypA can interact with MDA5 but they fail to follow up on these findings.

3) Furthermore, the mechanistic data (Figure 5 and Figure 6) heavily rely on overexpression experiments, and critical controls are missing in many experiments. Sometimes it is unclear from the figure labeling what was done in the experiment.

Major points that must be addressed:

1) Figure 1: The authors propose that CypA specifically promotes RIG-I signaling, and thus inhibits viruses sensed by RIG-I. However, they also provide data showing that CypA interacts with MDA5 (Figure 4). The authors need to also test the replication of viruses sensed specifically by MDA5, and test cytokine induction by these viruses.

2) CypA has been previously shown to act downstream of innate sensors, thereby promoting innate immune responses. Therefore, it is essential to show the relevance and contribution of the proposed CypA mechanisms, which are proposed to be at the level of RIG-I and MAVS, to IFN induction. The authors should test the effect of CypA overexpression and knockdown on virus replication and IFN induction in RIG-I or MAVS knockout cells. Moreover, the authors should examine the effect of CypA on RLR-independent signaling, such as the cGAS-STING pathway, which should not be affected.

3) Figure 2—figure supplement 4, panels C-D: In the spleen, levels of IFN-α and IFN-β increase upto 24h whereas levels of ISGs (which are believed to be downstream of IFN-α/β) decrease. The most significant difference in ISGs between WT and PPIA KO mice is seen at 6 h – a time that precedes a significant difference in IFNa and IFNb – and is lost by 24 h when the difference in IFN between WT and PPIA KO mice is most significant. This is in contrast to what is seen in the lung (Figure 2). Can the authors explain this discrepancy and address it in the Discussion?

4) The mechanism of action of CypA i.e. its effect on ubiquitination of RIG-I and MAVS is worked out only under overexpression conditions in 293 cells (Figure 5 and Figure 7). The authors need to show the effect of CypA knockdown on endogenous RIG-I K63-ubiquitination and RIG-I-TRIM25 binding as well as endogenous MAVS ubiquitination in infected cells. In addition, to relate this mechanism to the physiological effect of CypA on SeV-induced type 1 IFN (Figure 1 and Figure 2), it is important to show that endogenous CypA can trigger ubiquitination of endogenous RIG-I and MAVS upon SeV infection. One way to do this is by infecting WT and *Ppia^-/-^* BMDMs with SeV, followed by i.p. for either RIG-I or MAVS and blotting with K63- or K48-linkage specific polyubiquitin antibodies.

5) How do the authors integrate their data showing CypA promoting TRIM25 binding to RIG-I, but on the other hand, CypA competing with TRIM25 for MAVS binding. Additional experiments integrating the two mechanisms need to be performed.

6) The figure labeling and legends are not sufficiently described. For example, it is unclear which antibodies were used to detect the transfected proteins. In many figures the authors transfected tagged proteins, but then just label the blots with the name of the protein. Therefore, it is unclear which antibody (presumably the antibody against the tag) was used. The authors need to label the figures correctly, indicating which antibodies were used for each IP or western blot. Furthermore, this also needs to be described in the figure legend. In particular, in Figure 4, the amount of CypA in whole cells lysates in the rightmost lane is quite low compared to other samples. This lane (i.p: IRF3, blot: CypA) is an important specificity control to show that CypA specifically interacts with RIG-I, MDA-5 and MAVS which are involved in CypA-regulated production of type 1 IFN, and does not associate non-specifically with other RIG-I pathway components under overexpression conditions in 293 cells. The authors should replace this with an experiment where CypA levels in lysates are equivalent across samples. Figure 5, Figure 6 and Figure 7: A blot for HA in the lysates needs to be included to show equivalent transfection of HA-tagged Ub, K63-Ub and K68-Ub across samples.

7) Figure 4: In the confocal images, one expects MAVS to be primarily on the mitochondria but its staining appears to be homogeneous and a punctate distribution is not seen – it isn't clear why, but this could be due to antibody non-specificity or lack of adequate image resolution, both of which can confound a colocalization experiment. Similarly, it isn't clear that the cells being imaged are productively infected with SeV. If so, one would expect RIG-I and MDA-5 to also be at least partly mitochondrial / punctate in virus infected cells. To convincingly show by confocal imaging that CypA colocalizes with RIG-I, MDA-5 or MAVS after SeV infection, the authors should show representative images of CypA localization relative to that of RIG-I, MDA-5 and MAVS in uninfected cells.

8) Figure 6: It is unclear what this figure shows as it seems that no MAVS was co-transfected, or endogenous MAVS pulled down. The authors need to show also the WCLs showing HA-Ub expression.

9) Figure 7: CypA-MAVS interaction is reduced in presence of TRIM25, but TRIM25-MAVS interaction does not appear to be reduced in presence of CypA. So, although the authors present evidence that both CypA and TRIM25 interact with a similar stretch of MAVS i.e. aa 360-450 (Figure 7) and within this region aa K371 and K420 are the ubiquitination target sites for TRIM25 (Figure 7) as well as the sites that CypA targets to stabilize MAVS (Figure 6), the blot in Figure 7 does not show that CypA competitively inhibits TRIM25 interaction with MAVS. This could just be an issue of saturating exposure on the blot, or it is possible that CypA somehow inhibits the ubiquitin ligase activity of TRIM25 rather than its interaction with MAVS per se. The authors should clarify and either change the blot or reword the text (particularly in the second paragraph of the subsection “CypA stabilizes MAVS by inhibiting TRIM25-mediated K48-linked ubiquitination of MAVS”, but also in other places where a 'competitive interaction' is mentioned) accordingly.

[Editors' note: further revisions were requested prior to acceptance, as described below.]

Thank you for resubmitting your work entitled "Cyclophilin A-regulated ubiquitination is critical for RIG-I-mediated antiviral immune responses" for further consideration at *eLife*. Your revised article has been favorably evaluated by Michel Nussenzweig (Senior Editor), a Reviewing Editor, and two reviewers.

The manuscript has been improved but there are some remaining issues that need to be addressed before acceptance.

Please make the corrections / improvements to the figures as indicated below. The changes made will be reviewed only by the editor so the final decision on the submission will be made quickly after a new revised manuscript is received.

*Reviewer #1:*

The revised manuscript is significantly improved, and the authors' conclusions strengthened by the additional data. Below are listed two remaining issues:

1) The WCL HA-Ub blots in almost all experiments are overexposed, making it impossible to compare the signals between the different samples. The authors should include lower-exposure images for the HA blots in Figure 5, Figure 6, Figure 7.

2) Figure 7: It is unclear how the authors calculated the relative SeV M transcripts upon FLAG-RIG-I/MAVS overexpression, as a control sample with SeV infection + vector transfection is missing. The authors should explain this.

---

## [Author Response]

*Major points that must be addressed:*

*1) Figure 1: The authors propose that CypA specifically promotes RIG-I signaling, and thus inhibits viruses sensed by RIG-I. However, they also provide data showing that CypA interacts with MDA5 (Figure 4). The authors need to also test the replication of viruses sensed specifically by MDA5, and test cytokine induction by these viruses.*

We have examined the replication of EMCV and induction of IFN-β and ISGs response to EMCV in 293T/CypA+ and 293T/CypA+ by quantitative PCR. The results showed that CypA promoted EMCV-triggered MDA5 signaling and inhibited the replication of EMCV (Figure 1—figure supplement 3). We will further explore the molecular mechanism of CypA in MDA5 signaling in future. In this manuscript, we mainly focus on RIG-I-mediated antiviral immune responses.

*2) CypA has been previously shown to act downstream of innate sensors, thereby promoting innate immune responses. Therefore, it is essential to show the relevance and contribution of the proposed CypA mechanisms, which are proposed to be at the level of RIG-I and MAVS, to IFN induction. The authors should test the effect of CypA overexpression and knockdown on virus replication and IFN induction in RIG-I or MAVS knockout cells. Moreover, the authors should examine the effect of CypA on RLR-independent signaling, such as the cGAS-STING pathway, which should not be affected.*

a) CypA has been shown to interact with p65 and inhibit the ubiquitin-proteasome degradation of p65, thereby promoting innate immune responses (Sun et al., PloS ONE, 2014,9: e96211). Our data showed that CypA targeted the upstream RIG-I and MAVS to upregulate RIG-I-mediated signaling pathway. To explore the relevance and contribution of the proposed CypA mechanisms at the level of RIG-I and MAVS, we tested the effect of CypA on IFN-β induction in RIG-I knockout cells (293T/RIG-I^-/-^) transfected with Flag-RIG-I or Flag -MAVS, along with CypA siRNA or control siRNA. CypA increased IFN-β expression and IRF3/p65 phosphorylation with the treatment of overexpressed RIG-I and MAVS respectively (Figure 7). Then we further blocked the downstream p65 using NF-κB inhibitor (BAY-11-7082). We found that CypA still had impact on IFN-β expression and IRF3 phosphorylation when 293T/RIG-I^-/-^ were transfected with Flag-RIG-I and treated with BAY-11-7082, or when 293T/RIG-I^-/-^ were transfected with Flag-MAVS and treated with BAY-11-7082 (Figure 7). Collectively, these data indicate that CypA is able to promote type I IFN production at the level of RIG-I and MAVS, which is independent of the downstream p65. Accordingly, CypA could inhibit SeV replication via RIG-I- and MAVS-directed signaling pathways (Figure 7).

b) We have tested the effect of CypA on cGAS-STING pathway triggered by herpes simplex virus type 1 (HSV-1). The results showed that HSV-1-triggered cGAS-STING pathway could not be influenced by CypA (Figure 1—figure supplement 3).

*3) Figure 2—figure supplement 4, panels C-D: In the spleen, levels of IFN-α and IFN-β increase upto 24h whereas levels of ISGs (which are believed to be downstream of IFN-α/β) decrease. The most significant difference in ISGs between WT and PPIA KO mice is seen at 6 h – a time that precedes a significant difference in IFNa and IFNb – and is lost by 24 h when the difference in IFN between WT and PPIA KO mice is most significant. This is in contrast to what is seen in the lung (Figure 2). Can the authors explain this discrepancy and address it in the Discussion?*

There could be two main reasons for this discrepancy: First, the cell type and innate immune response in the spleen are quite different from those in the lung. It has been reported that the induction of ISGs is both tissue and time specific, and the expression profiles of ISGs (such as OAS, Mx-1 and PRK) triggered by IFN-α in the lung and spleen (Röll et al. BMC Genomics, 2017, 18:264) are similar with those of IFIT1, IFIT2 and Ccl5 in our studies. Second, ISGs expression in the spleen might be induced rapidly after SeV infection through an IFN-independent signaling pathway, and is functionally important for controlling viral replication before the onset of more robust and sustained IFN activation. A previous study has demonstrated that peroxisomal MAVS triggers an IFN-independent signaling pathway that promotes ISG expression after virus infection (Dixit E, et al. Cell. 2010; 141:668–681). In the present study, we found that CypA could inhibit proteasome degradation of MAVS. Therefore, it is possible that CypA might promote peroxisomal MAVS-mediated ISG expression. The exact mechanism need to be further studied.

*4) The mechanism of action of CypA i.e. its effect on ubiquitination of RIG-I and MAVS is worked out only under overexpression conditions in 293 cells (Figure 5 and Figure 7). The authors need to show the effect of CypA knockdown on endogenous RIG-I K63-ubiquitination and RIG-I-TRIM25 binding as well as endogenous MAVS ubiquitination in infected cells. In addition, to relate this mechanism to the physiological effect of CypA on SeV-induced type 1 IFN (Figure 1 and Figure 2), it is important to show that endogenous CypA can trigger ubiquitination of endogenous RIG-I and MAVS upon SeV infection. One way to do this is by infecting WT and Ppia^-/-^ BMDMs with SeV, followed by i.p. for either RIG-I or MAVS and blotting with K63- or K48-linkage specific polyubiquitin antibodies.*

The reviewers’ suggestion is greatly appreciated. We have further confirmed the effect of CypA on endogenous ubiquitination of RIG-I (Figure 5) and MAVS (Figure 7) as well as RIG-I-TRIM25 binding (Figure 5) in WT and *Ppia^-/-^* BMDMs infected by SeV.

*5) How do the authors integrate their data showing CypA promoting TRIM25 binding to RIG-I, but on the other hand, CypA competing with TRIM25 for MAVS binding. Additional experiments integrating the two mechanisms need to be performed.*

We are greatly appreciated for the reviewers’ suggestion to let us make an in-depth analysis of our results. CypA is widely distributed in almost all tissues. Multiple lines of evidence have revealed that CypA interacts with a large number of proteins and plays various biological roles through different mechanisms. We found that CypA-MAVS interaction was reduced in the presence of TRIM25, and TRIM25-MAVS interaction also appeared to be reduced in the presence of CypA. In addition, both CypA and TRIM25 interacted with a similar stretch of MAVS i.e. aa 360-450 and within this region aa K371 and K420 were the ubiquitination target sites for TRIM25 as well as the sites that CypA targeted to stabilize MAVS. All these results suggest that CypA competes with TRIM25 for MAVS binding to inhibit TRIM25-mediated K48-linked ubiquitination of MAVS. But on the other hand, we found that CypA promoted the interaction between TRIM25 and RIG-I, which is a quite different mechanism from that at the level of MAVS. As is well known, TRIM25 interacts with RIG-I-N to deliver the Lys 63-linked ubiquitin to the CARDs of RIG-I (Gack et al. Nature, 2007, 19;446(7138):916-920). To further explore the binding mechanisms of CypA, RIG-I and TRM25, we performed coimmunoprecipitation assays to determine the CypA-binding region of RIG-I. We found that CypA interacted with RIG-I-C, indicating that CypA and TRIM25 bind to the different domain of RIG-I respectively. We speculated that the conformation of RIG-I might be changed as soon as CypA interacted with RIG-I-C, then the binding site of TRIM25 was exposed, which facilitated the interaction between TRIM25 and RIG-I-N. The detail structural study is an interesting future direction.

*6) The figure labeling and legends are not sufficiently described. For example, it is unclear which antibodies were used to detect the transfected proteins. In many figures the authors transfected tagged proteins, but then just label the blots with the name of the protein. Therefore, it is unclear which antibody (presumably the antibody against the tag) was used. The authors need to label the figures correctly, indicating which antibodies were used for each IP or western blot. Furthermore, this also needs to be described in the figure legend. In particular, in Figure 4, the amount of CypA in whole cells lysates in the rightmost lane is quite low compared to other samples. This lane (i.p: IRF3, blot: CypA) is an important specificity control to show that CypA specifically interacts with RIG-I, MDA-5 and MAVS which are involved in CypA-regulated production of type 1 IFN, and does not associate non-specifically with other RIG-I pathway components under overexpression conditions in 293 cells. The authors should replace this with an experiment where CypA levels in lysates are equivalent across samples. Figure 5, Figure 6 and Figure 7: A blot for HA in the lysates needs to be included to show equivalent transfection of HA-tagged Ub, K63-Ub and K68-Ub across samples.*

a) We apologize for the confusion in figure labeling and legends. We have corrected the figure labeling and legends carefully.

b) We have replaced Figure 4 with an experiment where CypA levels in lysates are equivalent across samples.

c) The blot for HA in the lysates has been added.

*7) Figure 4: In the confocal images, one expects MAVS to be primarily on the mitochondria but its staining appears to be homogeneous and a punctate distribution is not seen – it isn't clear why, but this could be due to antibody non-specificity or lack of adequate image resolution, both of which can confound a colocalization experiment. Similarly, it isn't clear that the cells being imaged are productively infected with SeV. If so, one would expect RIG-I and MDA-5 to also be at least partly mitochondrial / punctate in virus infected cells. To convincingly show by confocal imaging that CypA colocalizes with RIG-I, MDA-5 or MAVS after SeV infection, the authors should show representative images of CypA localization relative to that of RIG-I, MDA-5 and MAVS in uninfected cells.*

Thanks for reviewers’ suggestion. We have changed the antibodies for confocal experiment and obtained better images (Figure 4). In addition, we have added representative images in uninfected cells as reviewers’ suggestion (Figure 4, upper).

*8) Figure 6: It is unclear what this figure shows as it seems that no MAVS was co-transfected, or endogenous MAVS pulled down. The authors need to show also the WCLs showing HA-Ub expression.*

We apologize for the confusion in figure labeling and legends. We have corrected them. A blot for HA-Ub expression in WCLs have been added.

*9) Figure 7: CypA-MAVS interaction is reduced in presence of TRIM25, but TRIM25-MAVS interaction does not appear to be reduced in presence of CypA. So, although the authors present evidence that both CypA and TRIM25 interact with a similar stretch of MAVS i.e. aa 360-450 (Figure 7) and within this region aa K371 and K420 are the ubiquitination target sites for TRIM25 (Figure 7) as well as the sites that CypA targets to stabilize MAVS (Figure 6), the blot in Figure 7 does not show that CypA competitively inhibits TRIM25 interaction with MAVS. This could just be an issue of saturating exposure on the blot, or it is possible that CypA somehow inhibits the ubiquitin ligase activity of TRIM25 rather than its interaction with MAVS per se. The authors should clarify and either change the blot or reword the text (particularly in the second paragraph of the subsection “CypA stabilizes MAVS by inhibiting TRIM25-mediated K48-linked ubiquitination of MAVS”, but also in other places where a 'competitive interaction' is mentioned) accordingly.*

We have repeated the coimmunoprecipitation experiment. CypA-MAVS interaction was reduced in the presence of TRIM25, and TRIM25-MAVS interaction also appeared to be reduced in the presence of CypA (Figure 7).

[Editors' note: further revisions were requested prior to acceptance, as described below.]

*Reviewer #1:*

*The revised manuscript is significantly improved, and the authors' conclusions strengthened by the additional data. Below are listed two remaining issues:*

*1) The WCL HA-Ub blots in almost all experiments are overexposed, making it impossible to compare the signals between the different samples. The authors should include lower-exposure images for the HA blots in Figure 5, Figure 6, Figure 7.*

Lower-exposure images for the HA blots have been shown in Figure 5, Figure 6, Figure 7.

*2) Figure 7: It is unclear how the authors calculated the relative SeV M transcripts upon FLAG-RIG-I/MAVS overexpression, as a control sample with SeV infection + vector transfection is missing. The authors should explain this.*

We agree with the reviewer’s comments. The control sample with empty vector transfection have been replaced to that with empty vector transfection and SeV infection. We also describe the control sample and the method used to calculate the fold induction in the figure legend.